# Sparser, Faster, Lighter Transformer Language Models

**Edoardo Cetin** [* 1] **Stefano Peluchetti** [* 1] **Emilio Castillo** [* 2] **Akira Naruse** [2] **Mana Murakami** [2] **Llion Jones** [1]

## Abstract

Scaling autoregressive large language models (LLMs) has driven unprecedented progress but comes with vast computational costs. In this work, we tackle these costs by leveraging unstructured sparsity within an LLM's feedforward layers, the components accounting for most of the model parameters and execution FLOPs. To achieve this, we introduce a new *sparse packing format* and a set of *CUDA kernels* designed to seamlessly integrate with the optimized execution pipelines of modern GPUs, enabling efficient sparse computation during LLM inference and training. To substantiate our gains, we provide a quantitative study of LLM sparsity, demonstrating that simple L1 regularization can induce over 99% sparsity with negligible impact on downstream performance. When paired with our kernels, we show that these sparsity levels translate into substantial throughput, energy efficiency, and memory usage benefits that increase with model scale. We will release all code and kernels under an open-source license to promote adoption and accelerate research toward establishing sparsity as a practical axis for improving the efficiency and scalability of modern foundation models [1].

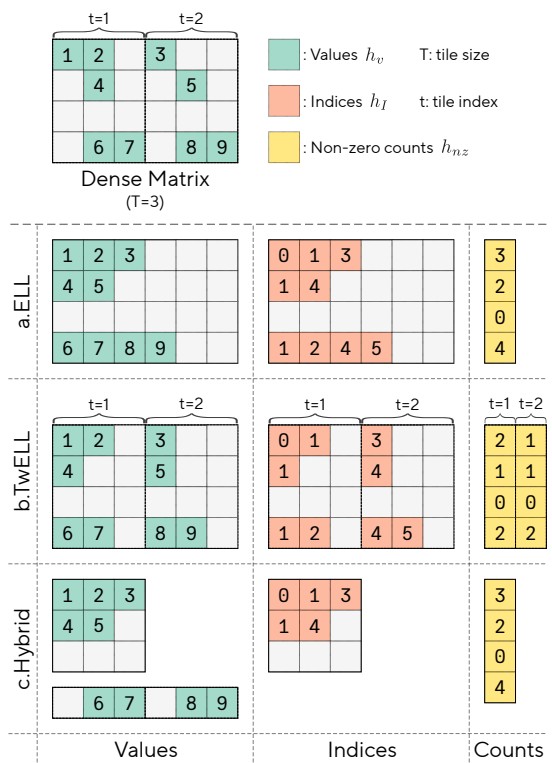

*Figure 1.* Comparison of ELL with our new TwELL and Hybrid sparse formats designed for LLM inference and training.

## 1. Introduction

Large Language Models (LLMs) have revolutionized natural language processing, demonstrating unprecedented capabilities in text generation, reasoning, and knowledge retrieval (OpenAI, 2023; Team et al., 2023). The core component driving these advancements has been massive computational investments into scaling the seminal Transformer architecture, with current LLMs reaching hundreds of billions of parameters (Vaswani et al., 2017; Radford et al., 2019; Brown et al., 2020). However, with increasingly larger models requiring vast computational resources for both inference and training, there is a growing need for fundamental efficiency improvements to ensure the present and future sustainability of the field (Schwartz et al., 2020; Luccioni et al., 2023).

One seminal avenue for improving the efficiency of machine learning models is sparsity (LeCun et al., 1989; Han et al., 2015; Hoefler et al., 2021). For modern overparameterized LLMs, recent investigations have even documented that sparsity arises naturally in their feed-forward layers, with only a small fraction of hidden neurons activated for any given token (Zhang et al., 2022b; Li et al., 2023). Thus, with feed-forward computation accounting for over two-thirds of the parameters and over 80% of the total FLOPs in larger models (Pope et al., 2023), sparsity seemingly offers a natural opportunity for concrete computational savings.

---

[*]Core contribution [1]Sakana AI [2]NVIDIA. Correspondence to: Edoardo Cetin <edo@sakana.ai>, Emilio Castillo <ecastillo@nvidia.com>.

*Proceedings of the 43rd International Conference on Machine Learning*, Seoul, South Korea. PMLR 306, 2026. Copyright 2026 by the author(s).

[1]Our open-source code and kernels are available at: github.com/SakanaAI/sparser-faster-llms

However, a frustrating paradox has blocked progress: despite performing far less theoretical computation, official kernels implementing sparse operations can often run slower than dense operations on modern GPUs. The culprit is a fundamental mismatch between unstructured sparsity and GPU architectures, whose hardware and software stacks have been heavily optimized for dense computation patterns (Lawson et al., 1979; NVIDIA, 2025a;b;c). In contrast, heterogeneous workloads together with the overheads from materializing and managing sparse indices have been critical challenges preventing generalized computational savings. Due to these challenges, previous attempts to realize efficiency gains have relied on considerable deviations from modern training recipes and have yet to see practical adoption (Liu et al., 2023; Wang et al., 2024).

In this work, we introduce new kernels designed for modern NVIDIA GPUs to bridge this gap and leverage unstructured sparsity to deliver substantial speedups while reducing memory requirements and energy consumption during both LLM inference and training. Our kernels build on Tilewise ELLPACK (TwELL), a new packing format for sparse data that can be naturally materialized in the epilogue of highly-optimized matrix multiplication kernels, removing a canonical bottleneck of prior packing schemes. Starting from TwELL, our inference kernels fuse multiple matrix multiplications into a single optimized pipeline that minimizes computation, while our training kernels further reduce the sparse representation to a hybrid format that trivializes the storage costs of intermediate activations.

To substantiate our gains, we provide a quantitative study of LLM sparsity across model scales, demonstrating that mild levels of L1 regularization can achieve over 99% sparsity with negligible impact on downstream performance. Through our new kernels, we show these sparsity levels translate into increasing benefits with larger parameter counts in terms of processing throughput, energy savings, and memory requirements – delivering up to 20.5% and 21.9% speedups in forward execution and training for models with billions of parameters. We analyze how these benefits specifically come from the computational unevenness across network layers and natural language data, which can be inherently leveraged in sparse models. By providing a clear demonstration of its practical benefits, we hope this work will help establish sparsity as a new axis for improving the scalability and performance of modern foundation models.

In summary, our main contributions are threefold:

1. We introduce and share new CUDA kernels for inference and training, with several key innovations to make sparse LLMs cheaper, faster, and lighter on modern GPUs.

2. We provide a quantitative analysis showing that high levels of unstructured sparsity can be achieved using mild L1 regularization with negligible compromises on performance.

3. We demonstrate and analyze how our kernels leverage such sparsity with substantial and increasing benefits at larger scales for LLMs with billions of parameters.

## 2. Large Language Models and Sparsity

The original transformer used a simple 2-layer feedforward block, which has seen considerable evolution since its conception (Vaswani et al., 2017). In recent architectures, the design of this module has largely converged on a gated extension that has consistently proven empirical superiority (Shazeer, 2020). While in this work we release kernels for both the original and gated blocks, we focus our main text on the newer design and defer to Appendix C for further discussions, results, and comparisons with the older variant.

### 2.1. Feed-forward Modules as Sparse Knowledge Stores

A modern gated feed-forward block (Shazeer, 2020) is parameterized by three weight matrices $W_g \in \mathbb{R}^{K \times N}$, $W_u \in \mathbb{R}^{K \times N}$, and $W_d \in \mathbb{R}^{N \times K}$ representing the gate, up, and down projection matrices, respectively. In our notation, we use $M$ to denote the feed-forward block's effective batch size over all batched sequences and positions, $K$ to denote its input/output dimensions, and $N$ to denote its expanded hidden dimension. The gate and up projection matrices both process the block's input batch $x \in \mathbb{R}^{M \times K}$ and produce the up and gate activations $h_g$ and $h_u \in \mathbb{R}^{M \times N}$, where the symmetry between the two is broken with a non-linear activation function $\sigma$. These projections are then combined with elementwise multiplication into a unified hidden representation $h \in \mathbb{R}^{M \times N}$, before being projected back to their original dimensionality using the down projection weights $W_d$, to compute the block's outputs $y \in \mathbb{R}^{M \times K}$:

$$h_u = xW_u, h_g = \sigma(xW_g), h = h_u \odot h_g, y = hW_d. \quad (1)$$

Since the hidden dimension $N$ is typically much larger than $K$, feed-forward blocks can often account for most of the model's parameters and FLOPs. We note that a common conceptualization of these architectural components is that of a *dynamic key-value memory* (Geva et al., 2021; Dai et al., 2022). In this mental model, the inner products between $x$ and the columns of $W_g$ and $W_u$ induce *keys* $h$, while the rows of $W_d$ are seen as *values* acting as memory slots that can be dynamically retrieved based on the input.

### 2.2. Simple Ingredients for Training Sparse LLMs

We employ a simple recipe to induce varying levels of sparsity in the feed-forward activations, making minimal devia-

tions from established architectures and training objectives. First, we use the ReLU as the activation function of choice following the gate projections. Second, we add a simple L1 loss to the standard cross-entropy with a tunable coefficient $L_1$ to promote sparsity across the model's $L$ layers:

$$L_1 \times \frac{1}{L} \sum_{l=1}^{L} \frac{1}{MN} \sum_{m=1}^{M} \sum_{n=1}^{N} |h^l[m,n]|. \tag{2}$$

We note that many recent LLM architectures have deviated from using ReLUs in favor of smoother activation functions such as SiLU, with minor but consistent benefits (Shazeer, 2020; Touvron et al., 2023; Yang et al., 2024). In Appendix C, we provide direct empirical comparisons between these choices and also refer to orthogonal studies in the recent literature showing that domain-specific performance differences can be bridged with targeted training techniques (Mirzadeh et al., 2023; Lomeli et al., 2025).

## 3. Making Sparse LLMs Fast

We introduce new CUDA kernels for inference and training that leverage unstructured sparsity to efficiently rework the computation in the feed-forward blocks of an LLM. The algorithms underlying our kernels build on TwELL, a new sparse format specifically designed for seamless kernel fusion to realize the inherent throughput and memory benefits of sparsity with minimal overheads. In this section, we describe the core components and advantages of our new kernels with algorithmic descriptions that summarize their logic at the level of individual cooperative thread arrays (CTAs). We refer to Appendix A for code listings and more detailed design discussions of the thread-level CUDA implementations for H100 GPUs.

### 3.1. Sparse Formats and Kernels

The ELLPACK format (ELL) is considered the state-of-the-art for fast and efficient sparse matmuls (Kincaid et al., 1989). This format was leveraged in some of the earliest GPU implementations of sparse algebra (Bell & Garland, 2009), with more recent work focused on developing packing and sorting variants for better performance (Kreutzer et al., 2014; Anzt et al., 2014). As shown in part a. of Figure 1, an $M \times N$ matrix $h$ in the ELL format is stored as two padded matrices $h_v$ and $h_I$ of size $M \times N_{nz}$ with the non-zero values of $h$ and their column indices packed at the beginning of each row. This format prioritizes downstream usability over storage, padding the rows up to the maximum number of nonzero elements $N_{nz}$ for efficient retrieval.

The main logic in most matmul kernels to perform $y = hW$ with ELL, is to launch different parallel accumulations for each row $m = 0, \ldots, M-1$ of the sparse matrix $h$ using a set number of threads. In each accumulation, the kernel

---

**Algorithm 1** *Gate projection with TwELL storage*

1: **Parameters:** Tile sizes $T_n, T_m$, compression ratio C
2: **Input:** Dense $x \in \mathbb{R}^{M \times K}$, $W_g \in \mathbb{R}^{K \times N}$,
3: **Output:** Sparse $h_v \in \mathbb{R}^{M \times N/C}$, $h_I \in \mathbb{N}^{M \times N/C}$, $h_{nz} \in \mathbb{N}^{M \times N_T}$
4: **for all** tiles at $(m_0, n_0)$ **in parallel across CTAs do**
5:     $S \leftarrow x[m_0:m_0+T_m, :] \, W_g[:, n_0:n_0+T_n]$
6:     **for** $r \leftarrow 0 \ldots T_m-1$ **do**
7:         $m \leftarrow m_0 + r$ {global row index}
8:         $z \leftarrow 0$ {running count of non-zeros in tile}
9:         **for** $c \leftarrow 0 \ldots T_n-1$ **do**
10:           **if** $(S[r,c] > 0)$ **then**
11:             $n \leftarrow n_0/C + z$ {TwELL column index}
12:             $h_I[m, n] \leftarrow n_0 + c$ {store non-zero index}
13:             $h_v[m, n] \leftarrow S[r,c]$ {store non-zero value}
14:             $z \leftarrow z + 1$ {increment non-zero count}
15:           **end if**
16:         **end for**
17:         $h_{nz}[m, n_0/T_n] \leftarrow z$ {store final non-zero count}
18:     **end for**
19: **end for**

---

iterates for $n = 0, ..., N_{nz} - 1$ times, loading each column index $i = h_I[m, n]$ and value $v = h_v[m, n]$ of $h$, and multiplying it with the $K-$dimensional row of the dense weight $W[i, :]$. The key advantage of this format is that only a fraction of the weight columns and input values need to be processed, skipping the remaining zeros. To further reduce data access and computation, some later extensions like ELLPACK-R (Vazquez et al., 2010) also store the number of non-zeros in each row in a separate vector $h_{nz}$.

### 3.2. Fast, Fused, Sparse Inference

An effective predominant design for modern kernel pipelines is to maximize operator fusion and avoid unnecessary global memory accesses in order to best leverage the high compute throughput of modern NVIDIA GPUs. To this end, in a gated feed-forward block where sparsity patterns are determined by the gate activations $h_g$, prior sparse formats such as ELL suffer a major drawback. In essence, representing $h_g$ with ELL requires first accessing all elements in every row to count, compare, and align the non-zero values and indices. However, existing matmul kernels for dense inputs rely on parallelizing computation across small 2D tiles $T_m \times T_n$ of the outputs, computed independently in separate CTAs. Thus, obtaining the gate activations directly in the ELL format from the non-sparse inputs cannot be done in the same kernel of $h_g = \text{ReLU}(xW)$ without introducing expensive synchronization among different CTAs. In contrast, launching a separate kernel to do the conversion inherently introduces non-trivial overheads that concretely limit attainable throughput gains of the whole computation.

---

**Algorithm 2** *Fused up and down projections from TwELL*

---

1: **Input:** Tile size $T_n$, sparse $h_g \in \mathbb{R}^{M \times N}$, $(h_I, h_{nz})$
2: **Input:** $x \in \mathbb{R}^{M \times K}$, $W_u \in \mathbb{R}^{M \times N}$, $W_d \in \mathbb{R}^{N \times K}$
3: **Output:** $y \in \mathbb{R}^{M \times K}$
4: **for all** $m \in \pi(0..M-1)$ **in parallel do**
5:    $x_m \leftarrow x[m, :]$; $y_m \leftarrow 0$
6:    **for** $b \leftarrow 0 \ldots N/T_n - 1$ **do**
7:       $p \leftarrow h_{nz}[m, b]$
8:       **for** $t \leftarrow 0 \ldots p - 1$ **do**
9:          $n \leftarrow bT_n + h_I[m, bT_n + t]$ {non-zero index}
10:         $w_u = W_u[:, n]$ {$n$-th column of $W_u$}
11:         $u \leftarrow (x_m \cdot w_u)$ {sparse $h_u[m, n]$ element}
12:         $w_d \leftarrow W_d[n, :]$ {$n$-th row of $W_u$}
13:         $y_m \leftarrow y_m + h_g[m, n] u\, w_d$
14:       **end for**
15:    **end for**
16:    $y[m, :] \leftarrow y_m$
17: **end for**

---

To address these limitations, we introduce Tile-wise ELL-PACK (TwELL). As illustrated in part b. of Figure 1, rather than focusing on whole rows, TwELL divides the columns of $h_g$ in groups of horizontal 1D tiles of size $T$. Within each group of columns, TwELL stores the non-zero values present and their indices in a local ELL-based packing format, with the data of each row aligned at the beginning of each horizontal tile. This results in two matrices containing locally aligned values $h_v \in R^{M \times N/C}$ and indices $h_I \in R^{M \times N/C}$, where $C$ is a *compression factor* chosen so that $T/C$ is higher than the maximum number of non-zeros in any tile to avoid storage overflow. In our implementation of TwELL, we also store an additional matrix with the number of non-zero elements $h_{nz} \in R^{M \times N_T}$ to facilitate further computations, with as many columns as total tiles $N_T = \lceil N/T \rceil$. While inherently less expensive to derive, the main advantage of TwELL over ELL is actually ease of materialization following a modern tiled matmul: by setting the horizontal tiling dimensions to match, $T = T_n$, the TwELL format can be recovered in the same kernel performing $h_g = \text{ReLU}(xW)$ before storing the outputs to DRAM. Fusing the two operations removes the requirement of performing additional kernel spawns, memory reads, or synchronization steps, leading to a natural integration into existing LLM pipelines.

### 3.3. Kernels for TwELL Construction and Fast, Fused Inference

In Algorithm 1, we provide pseudocode to summarize the logic of our CUDA matmul kernel storing the sparse outputs in the TwELL format (lines 6-18). Given the output distribution patterns of tensor core operations, we obtain the memory addresses to store the packed non-zero values

$h_v$ and their indices $h_I$ by keeping a local non-zero count that only requires warp-level synchronization. While not an inherent requirement of TwELL, storing the number of non-zeros in each tile, $h_{nz}$, allows us to forego the overhead from initializing $h_I$ with any "padding" value and from the additional control logic of checking validity in future usages. While omitted from Algorithm 1, we leverage fast asynchronous TMA reads and writes by first caching the dense inputs and sparse TwELL outputs to shared memory. We also pipeline computation and global memory accesses with a persistent cooperative design similar to the one in CUTLASS (NVIDIA, 2025c).

For inference, we introduce a single additional kernel to perform the rest of the computation in the feedforward block, leveraging the gate activations stored in the TwELL format to efficiently *fuse* the up and down projections together. This kernel, summarized in Algorithm 2, is launched on a grid made of single warp CTAs each processing a different row $m$ of the input activations $x$. Minimizing the size of each CTA serves the purpose of maximizing concurrency and L2-hits across the grid, as non-zero activations tend to have high correlation within input sequences. The fused matmuls are executed by traversing over the sparsified activations with two nested for loops: the first one statically-unrolled over the number of column tiles (line 6) and the second one dynamically iterating over the corresponding number of non-zeros in each tile (line 8). For each non-zero activation at index $n$, the CTA collectively loads the $n^{th}$ column of $W_u$ and row of $W_d$ to perform a dot product, followed by a scalar-vector product, and accumulates its results (lines 9-13) – corresponding to the following computation:

$$y[m, :] = \sum_{n \in I_{nz}} \underbrace{h_g[m, n]}_{h_g \text{ non-zero}} \underbrace{(x[m, :] \cdot W_u[:, n])}_{h_u \text{ element}} \underbrace{W_d[n, :]}_{W_d \text{ row}}. \quad (3)$$

Implicitly materializing the $h_u$ values only inside the kernel serves to further reduce DRAM access to maximize throughput. Together, the kernels in our inference pipeline align the core principles of tiling and operator fusion into a single execution flow, harnessing the computational advantages of sparsity while minimizing its canonical overheads.

### 3.4. Hybrid Conversion for Efficient Storage

During training, memory becomes a key bottleneck for throughput as large intermediate activations and optimizer states are needed for backpropagation. Here, sparsity provides a natural opportunity to tackle these bottlenecks by trivializing intermediate storage costs and accelerating gradient computations. However, directly using TwELL with a high compression ratio or other ELL-based formats for this purpose inherently relies on the maximum number of non-zeros $N_{nz}$ to be known ahead of time and strictly small. However, as we will illustrate in Section 4, we find that these conditions are practically never met during LLM training as

**Algorithm 3** *Hybrid-to-dense matmul*

1: **Input:** $T_m, T_k, T_n, h := (h^s, h^d, h_I, h_b), W \in \mathbb{R}^{K \times N}$
2: **Output:** $y \in \mathbb{R}^{M \times N}$
3: $\pi_s \leftarrow \{m : h_b[m] = 0\}$; $\pi_d \leftarrow \{m : h_b[m] = 1\}$
4: **for all** $m_s \in 0..M^s-1$ **in parallel do**
5: $\quad m \leftarrow \pi_s[m_s]$ {global row index}
6: $\quad y_m \leftarrow 0$ {row accumulator}
7: $\quad$ **for** $j \leftarrow 0 \dots N_{\hat{n}z}-1$ **do**
8: $\quad\quad n \leftarrow h_I[m_s, j]$ {non-zero column index}
9: $\quad\quad v \leftarrow h^s[m_s, j]$ {sparse value}
10: $\quad\quad y_m \leftarrow y_m + v \cdot W[n, :]$ {sparse row update}
11: $\quad$ **end for**
12: $\quad y[m, :] \leftarrow y_m$
13: **end for**
14: **for all** tiles starting at $(m_0, n_0)$ **in parallel do**
15: $\quad S \leftarrow h^d[m_0:m_0+T_m, :]\, W[:, n_0:n_0+T_n]$
16: $\quad y[\pi_d[m_0:m_0+T_m], n_0:n_0+T_n] \leftarrow S$
17: **end for**

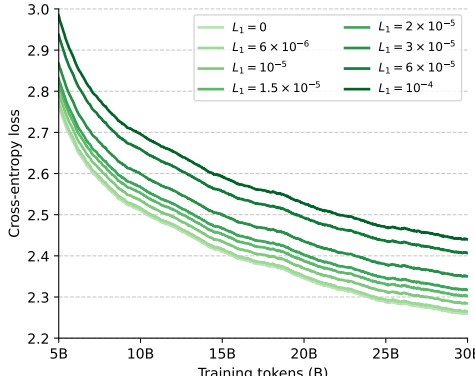

*Figure 2.* Training curves of LLMs across L1 regularization levels.

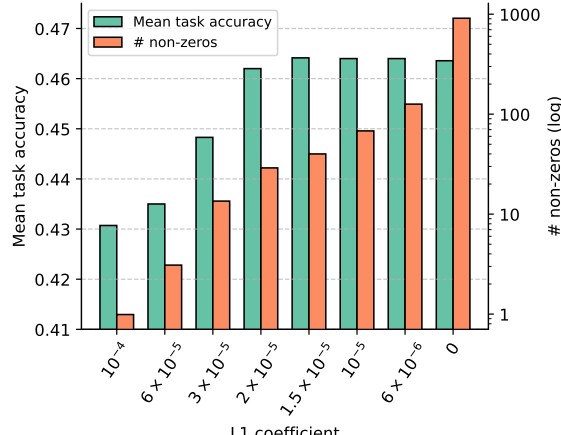

*Figure 3.* Downstream accuracy and sparsity statistics of LLMs across L1 regularization levels.

sparsity patterns exhibit significant non-uniformity across different tokens, with the maximum number of non-zeros often orders of magnitude larger than the average.

We overcome these limitations by first converting the TwELL activations to a *hybrid* sparse format and introducing a new set of custom kernels designed specifically for memory-efficient training. As illustrated part c. of Figure 1, our format dynamically partitions and stores the rows of $h_g$ either in an aggressively compact ELL matrix $h_g^s \in R^{M^s \times N_{\hat{n}z}}$ or a dense backup $h_g^d \in R^{M^d \times N}$. The partitioning logic simply routes the rows of $h_g$ based on their non-zero counts, which are cheaply computed from the locally aligned TwELL tiles. Our hybrid format also maintains a lightweight array of column indices $h_I \in R^{M^s \times N_{\hat{n}z}}$ matching the size of the sparsified ELL matrix, and a simple binary vector indicating the storage location of each row $h_b \in R^M$. In practice, we find that we can set $N_{\hat{n}z}$ over an order-of-magnitude lower than $N$ with minimal overflow into $h_g^d$, avoiding stringent ELL requirements while still trivializing memory and computation during the rest of the training step.

### 3.5. Kernels for Lightweight Efficient Training

After materializing $h_g$ in our hybrid format from the pre-activations $xW_g$, we design custom kernels to perform efficient hybrid-to-dense and dense-to-hybrid matmuls. We directly use these kernels to execute the rest of the forward pass, computing $h_u = xW_u$ and $y = hW_d$. Unlike inference, during training we execute the up and down projections in separate steps, allowing us to efficiently store the sparsified hidden states and minimize recomputation in the backward pass. In Algorithm 3, we outline the logic

of the hybrid-to-dense matmul for a generic input $h$ and weight $W$, with the dense-to-hybrid variant also following the same general structure. Our approach combines a typical ELL kernel, with each CTA processing individual rows of the output $y$ (lines 4-13), and a traditional tiled kernel using Tensor cores for the dense backup rows (lines 14-17). During the sparse portion of the matmul computation, we opt to statically-unroll the accumulation up to the maximum number of non-zeros $N_{\hat{n}z}$ for each row. Moreover, we also statically pre-allocate the dense backup portions of all the activations based on the sparsity statistics observed during training. We note that these design choices introduce minimal extra computation and storage costs, which are largely offset by avoiding dynamic overheads.

During the backward pass, we retrieve the sparsified activations together with the L1 and output gradients $\nabla y$, allowing us to backpropagate *without performing any expensive dense computation*. This is achieved using two additional kernels that support efficient injection of L1 gradients into a given sparsity pattern and efficient transposition of our hybrid format for future coalesced accesses. We first use the stored sparsity pattern of $h$ to obtain its gradients via our efficient dense-to-hybrid matmul $\nabla y W_d^T$, followed by the

L1 injection. With $\nabla h$ available, we recover the rest of the input and weight gradients with direct applications of our hybrid-to-dense and transposed kernels:

$$\nabla h_u = \nabla h \odot h_g, \nabla h_g = \nabla h \odot h_u,$$

$$\nabla W_u = x^\top \nabla h_u, \nabla W_g = x^\top \nabla h_g, \nabla W_d = h^\top \nabla y, \quad (4)$$

$$\nabla x = \nabla h_u W_u^\top + \nabla h_g W_g^\top.$$

Crucially, our execution logic reflects a deliberate design choice: rather than aggressively fusing individual operators, the training kernel pipeline is structured around the training step in its entirety. In this setting, the hybrid format minimizes backward computation and memory overheads, allowing us to avoid dense calculations while remaining robust to the highly non-uniform sparsity patterns that make ELL-based approaches traditionally brittle.

## 4. Experimental Results

### 4.1. Training and Evaluation Settings

We provide quantitative results evaluating the performance and efficiency of LLMs at different sparsity levels and scales. Our models are based on the "Transformer++" architecture, common to recent LLMs such as Qwen and Llama (Touvron et al., 2023; Yang et al., 2024) with the gated feedforward blocks described in Section 2. We train our models just above the chinchilla-optimal number of tokens for each model size (Hoffmann et al., 2022), using the fineweb dataset (Penedo et al., 2024). We default to a context length of 2048, a batch size of 1M tokens, and the AdamW optimizer with a weight decay of 0.1 and a cosine schedule (Loshchilov & Hutter, 2017). Other hyperparameters, such as the hidden size and total number of layers, are based on the model size and chosen based on modern practices (Hoffmann et al., 2022). We note that our sparse models use the same training hyperparameters as our non-sparse baselines, as the addition of L1 regularization in the feedforward blocks did not seem to affect other choices.

To measure model performance, we use cross-entropy scores and seven different common downstream tasks assessing logic and reasoning capabilities after pretraining (Clark et al., 2018; Zellers et al., 2019; Mihaylov et al., 2018; Bisk et al., 2020; Sakaguchi et al., 2021; Talmor et al., 2019). In this Section, we focus on aggregated performance metrics for conciseness, and we refer to Appendix D for the full per-task breakdowns. To measure efficiency, we analyze the throughput gains at different sparsity levels when integrating our training and inference kernels by recording execution times, memory requirements, and energy consumption at each stage. Across our experiments, we keep a fixed sequence length of 2048 and vary the micro batch size based on the available memory. Unless otherwise specified, we train and collect our results on a single node of eight H100 PCIe GPUs, a commonly available infrastructure in current

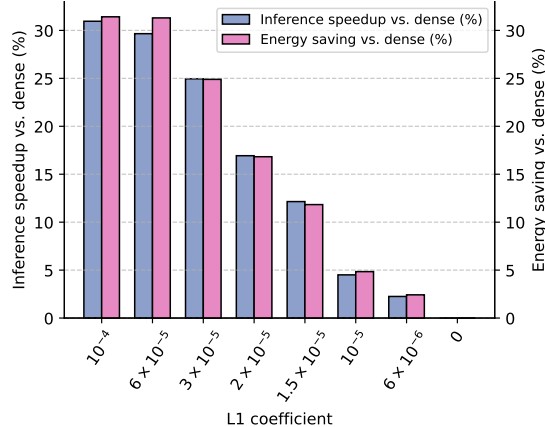

*Figure 4.* Forward pass speedups and energy savings from our sparse LLM inference kernels across L1 regularization levels. We note that the

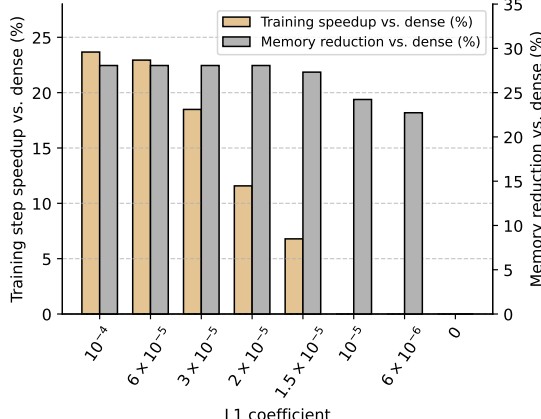

*Figure 5.* Training speedups and peak memory reduction from our sparse LLM training kernels across L1 regularization levels.

listings of cloud providers and scientific clusters. We refer to Appendix B for further details on our training and evaluation settings, together with full hyperparameters specific to each of the considered model sizes.

### 4.2. More Efficient LLMs with Unstructured Sparsity

**Performance across sparsity levels.** We start by evaluating the effect of introducing different levels of L1 regularization on the performance and sparsity of a 1.5B parameter model. In particular, we consider eight values for the $L_1$ coefficient, ranging from no regularization ($L_1 = 0$) to the point where less than a single neuron on average remains activated after training ($L_1 = 1 \times 10^{-4}$). In Figure 2, we show the training curves of the different models, while in Figure 3 we report downstream task performance together with the final number of non-zero activations averaged across the feed-forward blocks. While our 1.5B model has a hidden feedforward dimensionality of 5632, we find that the non-regularized model already attains more than 20% sparsity with only 911 neurons activated. Moreover, consistently with Mirzadeh

*Table 1.* Comparison of performance and efficiency statistics of sparse LLMs leveraging our kernels with traditional models. We note that the training speedups of the 2B model were recorded with a larger micro-batch size possible through the memory savings of our training kernels.

| Model scale | Sparse | Mean task accuracy | Forward execution (input tokens/ms) | Energy per token (mJ) | Training step (input token/ms) | Peak memory (GB) |
|---|---|---|---|---|---|---|
| 0.5B params 10B tokens | ✗ | 40.4% | 410 (0.0%) | 1.63 (0.0%) | 97.3 (0.0%) | 26.2 (0.0%) |
| | ✓ | 40.4% | 480 (+17.0%) | 1.43 (-11.8%) | 95.9 (-1.5%) | 21.2 (-19.2%) |
| 1B params 20B tokens | ✗ | 44.6% | 185 (0.0%) | 3.71 (0.0%) | 48.6 (0.0%) | 44.5 (0.0%) |
| | ✓ | 44.7% | 219 (+18.1%) | 3.17 (-14.6%) | 52.1 (+7.1%) | 33.1 (-25.5%) |
| 1.5B params 30B tokens | ✗ | 46.4% | 119 (0.0%) | 5.73 (0.0%) | 31.8 (0.0%) | 62.8 (0.0%) |
| | ✓ | 46.2% | 141 (+18.8%) | 4.87 (-15.0%) | 35.5 (+11.6%) | 45.1 (-28.1%) |
| 2B params 40B tokens | ✗ | 49.1% | 87.8 (0.0%) | 7.85 (0.0%) | 22.4 (0.0%) | 46.7 (0.0%) |
| | ✓ | 48.8% | 106 (+20.5%) | 6.51 (-17.0%) | 27.3 (+21.9%) | 57.1 (+22.3%) |

et al. (2023), we find that introducing small levels of regularization already pushes the average number of non-zeros orders of magnitude lower but with high variations across different tokens and layers. In particular, even at the highest regularization point, we find that a small fraction of tokens still excite several hundred neurons, indicating a reallocation of capacity. Despite this adaptivity, performance-wise, we do start seeing some performance degradation below 0.5% of activated neurons. Nonetheless, our results suggest that smaller levels of regularization do not visibly hinder capacity beyond the weight decay already induced by the AdamW optimizer: up until $L_1 = 3 \times 10^{-5}$, we record essentially no drop in task performance and a negligible increase of final cross-entropy within 2% of the unregularized baseline.

**Leveraging sparsity for faster and lighter LLMs.** We contrast our performance results by analyzing the efficiency improvements from integrating our kernels at different sparsity levels. In Figure 4, we provide the average relative speedups and total energy savings recorded during forward execution through our LLMs. Across all considered sparsity levels above $L_1 = 0$, we find that our inference kernels lead to visible throughput gains ranging up to 30%. These throughput gains are compounded by nearly 3% less GPU power draw above $L_1 = 3 \times 10^{-5}$, resulting in even higher energy savings. In Figure 5, we also show the average relative speedups and peak memory reduction with our training kernel. In line with our inference kernels, the speedups recorded throughout training significantly increase up to 24% with sparser models. Furthermore, the peak GPU memory required for training decreases by more than 24% even for the lowest considered sparsity level, reducing hardware barriers for efficient training at billion-parameter scales (we refer to Appendix D for results on an RTX6000). Taken together, we believe our results provide compelling evidence that specialized targeted kernels can make sparsity a new viable axis for the design of modern LLMs, leading to significant efficiency improvements across their full lifecycle.

**Sparsity across model scales**. We analyze how model scale

affects the performance and efficiency of sparse LLMs. For this analysis, we set $L_1 = 2 \times 10^{-5}$ based on our earlier results on the 1.5B model, which we recommend as a conservative threshold to avoid any significant performance degradation. In Table 1, we compare the performance and efficiency of sparse and non-sparse LLMs at the chinchilla-optimality boundary – ranging from a 0.5B model trained on 10B tokens to a 2B model trained on 40B tokens. Consistent with our earlier results, we find no performance drops beyond random deviations for all scales when mild L1 regularization is introduced. Furthermore, we find that LLMs become increasingly effective at supporting sparsity at larger scales, resulting in a lower number of average non-zero elements (from 39 to 24, going from the 0.5B to the 2B model). In turn, this makes all the aforementioned throughput and memory benefits of our kernels grow, with the 2B sparse model processing tokens 20.5% faster during inference and training 21.9% more efficiently. We note this latter training result was also made possible by the increased memory savings of our training kernels, which allowed us to fit a larger micro-batch size. These findings suggest that sparsity aligns well with recently prevailing scaling trends, highlighting its growing potential relevance for future model development.

### 4.3. The Properties of Sparse Large Language Models

We analyze how LLMs effectively allocate sparsity across their layers and batched samples. For our analysis, we collect the activations from $2^{20}$ input tokens using our 1.5B model trained with the suggested performance-preserving $L_1 = 2 \times 10^{-5}$. We complement this subsection with additional results in Appendix D, looking at additional levels of L1 regularization together with how sparsity evolves throughout training and its effects on dead neurons.

**Sparsity and model depth**. Figure 6 examines activations across model depth, relating the non-zero statistics of each layer to its respective contribution to inference speed-ups. While the average non-zeros across all layers is less than 30, the figure highlights clear variations in sparsity both across

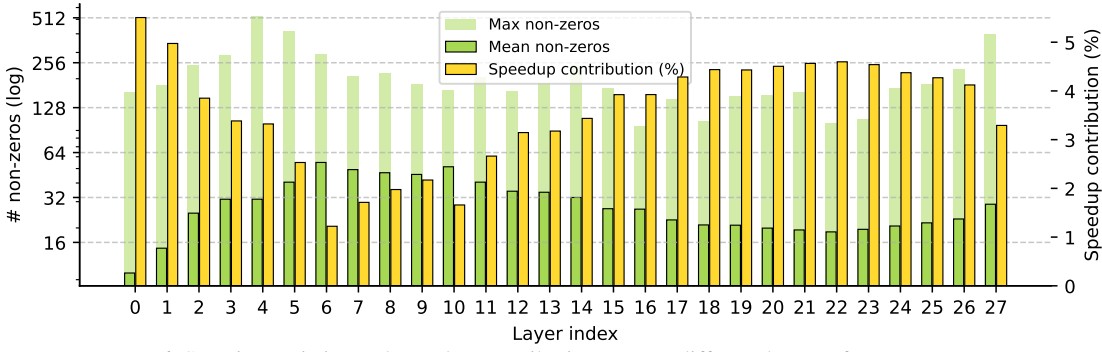

*Figure 6.* Sparsity statistics and speedup contributions across different layers of our sparse LLMs.

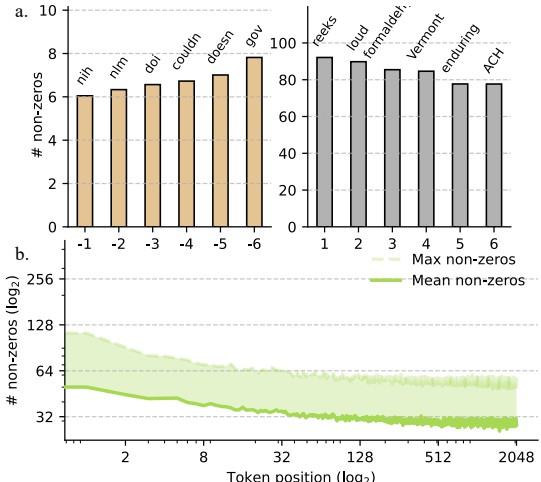

*Figure 7.* Sparsity statistics across LLM input tokens and positions.

and within individual layers. In particular, the first two layers are the least active, followed by a pronounced hump in the average number of non-zeros across the first half of the network. This sparsity pattern, peaking during early-middle layers, appears consistent with prior work suggesting that a substantial portion of an LLM's reasoning and knowledge retrieval occurs precisely at these depths (Wendler et al., 2024). Furthermore, within each layer, the maximum number of non-zeros often exceeds the layer's mean by more than an order of magnitude and shows no consistent pattern across the architecture. We also observe an intuitive and pronounced inverse correlation between each layer's average non-zeros and its relative speed-up, with a Pearson coefficient of over -0.996. In contrast, the maximum activation counts have a more limited effect on inference speedups, only noticeably in layer 8. This robustness is due to our kernel design, which hides the latency of highly activated tokens through maximally parallelized execution.

**Sparsity and input properties**. Given the high level of unevenness across activations, we analyze what inputs spur the peaks and troughs in non-zero activation counts. In part a. of Figure 7, we identify common tokens with the six lowest and highest average number of non-zeros, filtering out outliers occurring at a lower frequency than $1/2^{14}$. We

find that the tokens with the lowest non-zero activity often represent parts of common web links (*doi*, *nlm*, *gov*, *nih*) or contractions (*doesn*, *couldn*) that precede predictable next tokens in crawled web corpora. In contrast, tokens providing important contextual information about a passage have the highest activity, including particular verbs (*loud*, *enduring*) or nouns representing specific locations or substances (*Vermont*, *Greeks*, *formaldehyde*, *ACH"*). In part b. of Figure 7, we then plot how the average non-zeros vary with token position in the input sequence on a log-log scale. Interestingly, we find that the LLM allocates a much greater number of non-zeros to the very first tokens in a sequence, with an exponential decrease thereafter. Intuitively, these results indicate that LLMs appear to effectively focus their computational efforts on tokens with high information content and sequence positions where contextual cues from prior tokens are missing. Here, introducing sparsity not only provides an interpretable lens on model behavior, but also enables our kernels to leverage this inherent information unevenness for significant training and inference speedups.

## 5. Related Work

The emergence of sparsity in LLMs with ReLU activations and its theoretical benefits have been repeatedly documented in earlier work (Li et al., 2023; Mirzadeh et al., 2023). Since then, more recent methods have been proposed to enhance sparsity by altering modern gated architectures, reporting speedups on memory-bound GEMV operations when running sparse feed-forward layers in isolation on older generations of devices. TurboSparse (Song et al., 2024) studies boosting sparsity via repeated ReLU non-linearities, while ProSparse (Song et al., 2025) finetunes pretrained models by manually thresholding activations. Other work instead focused on introducing sparsity post-training, such as by predicting (Liu et al., 2023) and pruning activations to accelerate computation (Lee et al., 2024; Liu et al., 2024). In contrast to these prior works, our paper introduces *general-purpose kernels for compute-bound GEMM operations* in batched settings, where dense baselines on modern devices can execute up to orders-of-magnitude higher FLOP/s with large tiles and Tensor Cores. Moreover, focusing on the

batched setting allows our work to demonstrate that leveraging unstructured sparsity can provide empirical efficiency benefits during both LLM *inference and training*. We refer to Appendix E for an extended overview of prior work that more fundamentally reshapes architecture design.

## 6. Discussion and Future Work

In this work, we leverage unstructured sparsity to lessen the computational burdens of modern LLMs. For inference, we design a new sparse format and fused operations to efficiently execute the whole gated feed-forward blocks in only two kernel launches, minimizing global memory accesses and computation. For training, we introduce a new hybrid algorithm that dynamically schedules computation on both CUDA and Tensor cores, while trivializing storage costs of intermediate activations for backpropagation. We demonstrate that mild L1 regularization induces considerable levels of sparsity with negligible impact on downstream performance – which our kernels translate into significant gains in throughput, energy efficiency, and memory footprint at billion-parameter scales. While our work serves to provide a concrete demonstration of the benefits of sparse LLMs, there are numerous exciting avenues for future extensions. For instance, in Appendix C, we provide preliminary results indicating that the performance of highly sparse LLMs can be further improved with strategies targeted at dead-neuron mitigation. Moreover, fine-tuning existing dense models via recent sparsification approaches (Mirzadeh et al., 2023; Song et al., 2025) would allow bringing the benefits of our kernels to the vast library of pretrained LLMs available in the wild. By sharing our kernels, we hope our work will help promote sparsity as a new design axis to leverage for efficiency, ultimately reducing the growing energy and hardware costs of large-scale foundation models.

## Acknowledgments

**Edoardo Cetin** conceived the TwELL format, led the implementation and design of the CUDA kernels using TwELL, led model training and benchmarking, and made contributions to writing.

**Stefano Peluchetti** did early work on sparse model training, advised the project, and made contributions to writing.

**Emilio Castillo** conceived the hybrid format, co-led the implementation of the CUDA kernels and designed the training extensions, made contributions to kernel benchmarking, and made contributions to writing.

**Akira Naruse** advised the project, was involved in early discussions about method design, and worked on early implementations of the sparse kernels.

**Mana Murakami** was involved in early discussions about method design.

**Llion Jones** did initial explorations of sparse model training, advised the project, was involved in early discussions about method design, and made contributions to writing.

The authors would like to thank Takuya Akiba, Yujin Tang, and David Ha for providing valuable discussion and feedback throughout the project.

## Impact Statement

This paper presents work whose goal is to advance the field of Machine Learning. There are many potential societal consequences of our work, none which we feel must be specifically highlighted here.

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

# A. Kernels Implementation Details

## A.1. Inference Kernels Selection

```
1  template <const int T_m, const int T_n, const int T_k>
2  struct Tiles
3  {
4      alignas(128) __nv_bfloat16 a[T_m][T_k];
5      alignas(128) __nv_bfloat16 b[T_n][T_k];
6  };
7
8  template <
9      const int T_m,
10     const int T_n,
11     const int T_k,
12     const int QUEUE_SIZE,
13     const int T_n_compressed,
14     int PADDING = 4
15 >
16 struct SmemStorage
17 {
18     Tiles<T_m, T_n, T_k> queue[QUEUE_SIZE];
19     alignas(128) uint32_t c_packed[T_m][T_n_compressed + PADDING];
20 };
21
22 template <
23     const int T_m,
24     const int T_n,
25     const int T_k,
26     const int CLUSTER_DIM_m,
27     const int CLUSTER_DIM_n,
28     const int QUEUE_SIZE,
29     const int NUM_ACTIVE_SMs,
30     const int T_n_compressed,
31     const bool LOOP_OVERFLOW_STORAGE
32 >
33 __global__ __launch_bounds__(NUM_THREADS_PER_BLOCK)
34          __cluster_dims__(CLUSTER_DIM_m * CLUSTER_DIM_n, 1, 1)
35 void mm_wgmma_nt_kernel(
36     const CUtensorMap __grid_constant__ A_tm,
37     const CUtensorMap __grid_constant__ B_tm,
38     const CUtensorMap __grid_constant__ C_packed_tm,
39     const int* schedule_gmem_ptr,
40     const int schedule_size_per_sm,
41     const int K
42 )
43 {
44     static_assert(
45         (T_m == 64 * 2),
46         "Only T_m == 128 supported"
47     );
48
49     constexpr int CLUSTER_SIZE = CLUSTER_DIM_m * CLUSTER_DIM_n;
50     extern __shared__ __align__(1024) unsigned char dynamic_smem[];
51
52     int cluster_idx;
53     asm ("mov.u32 %0, %clusterid.x;\n" : "=r"(cluster_idx) :);
54
55     int cluster_lane_m;
56     asm volatile("mov.u32 %0, %cluster_ctarank;\n" : "=r"(cluster_lane_m) :);
57
58     int cluster_lane_n = cluster_lane_m % CLUSTER_DIM_n;
59     cluster_lane_m /= CLUSTER_DIM_n;
60
61     auto& tiles_s =
62         *reinterpret_cast<
63             SmemStorage<T_m, T_n, T_k, QUEUE_SIZE, T_n_compressed>*
64         >(dynamic_smem);
65     int* schedule_s = reinterpret_cast<int*>(
66         dynamic_smem
67         + sizeof(SmemStorage<T_m, T_n, T_k, QUEUE_SIZE, T_n_compressed>)
68     );
69
70     schedule_gmem_ptr += cluster_idx * schedule_size_per_sm;
71     if (threadIdx.x < schedule_size_per_sm) {
72         schedule_s[threadIdx.x] = schedule_gmem_ptr[threadIdx.x];
73     }
74
75     __syncthreads();
76
```

```
77      __shared__ __align__(8) uint64_t queue_full[QUEUE_SIZE];
78      __shared__ __align__(8) uint64_t queue_empty[QUEUE_SIZE];
79
80      if (threadIdx.x == 0) {
81          #pragma unroll
82          for (int queue_idx = 0; queue_idx < QUEUE_SIZE; ++queue_idx) {
83              ptx_init_smem_barrier(&queue_full[queue_idx], 1);
84              ptx_init_smem_barrier(&queue_empty[queue_idx], 2 * CLUSTER_SIZE);
85          }
86      }
87
88      asm volatile("barrier.cluster.arrive;\n" : :);
89      asm volatile("barrier.cluster.wait;\n" : :);
90
91      if (threadIdx.x < WARP_GROUP_SIZE) {
92          asm volatile("setmaxnreg.dec.sync.aligned.u32 %0;" :: "n"(24): "memory");
93
94          if (threadIdx.x == 0) {
95              int queue_idx = 0;
96              int queue_phase = 0;
97              uint16_t mask_multicast_m = 0;
98
99              if constexpr (CLUSTER_DIM_m > 1) {
100                 for (int i = 0; i < CLUSTER_DIM_m; ++i) {
101                     mask_multicast_m |= (1u << (i * CLUSTER_DIM_n));
102                 }
103                 mask_multicast_m <<= cluster_lane_n;
104             }
105
106             uint16_t mask_multicast_n;
107             if constexpr (CLUSTER_DIM_n > 1) {
108                 mask_multicast_n =
109                     ((1u << CLUSTER_DIM_n) - 1)
110                     << (cluster_lane_m * CLUSTER_DIM_n);
111             }
112
113             for (int schedule_it = 0;
114                  schedule_it < schedule_size_per_sm;
115                  ++schedule_it) {
116                 const int packed_tile = schedule_s[schedule_it];
117                 if (packed_tile == -1) {
118                     break;
119                 }
120
121                 int tile_coord_m = packed_tile >> 16;
122                 int tile_coord_n = packed_tile & 0xFFFF;
123
124                 if constexpr (CLUSTER_DIM_n > 1) {
125                     tile_coord_n *= CLUSTER_DIM_n;
126                     tile_coord_n += cluster_lane_n;
127                 }
128                 if constexpr (CLUSTER_DIM_m > 1) {
129                     tile_coord_m *= CLUSTER_DIM_m;
130                     tile_coord_m += cluster_lane_m;
131                 }
132
133                 for (int tile_start_k = 0;
134                      tile_start_k < K;
135                      tile_start_k += T_k, ++queue_idx) {
136                     if (queue_idx == QUEUE_SIZE) {
137                         queue_idx = 0;
138                         queue_phase ^= 1;
139                     }
140
141                     ptx_wait_barrier(&queue_empty[queue_idx], queue_phase);
142                     ptx_arrive_tx_smem_barrier(
143                         &queue_full[queue_idx],
144                         sizeof(tiles_s.queue[queue_idx].a)
145                         + sizeof(tiles_s.queue[queue_idx].b)
146                     );
147
148                     if constexpr (CLUSTER_DIM_n > 1) {
149                         if (cluster_lane_n == 0) {
150                             ptx_load_tile_tma_multicast_2d(
151                                 &tiles_s.queue[queue_idx].a[0][0],
152                                 &A_tm,
153                                 tile_coord_m * T_m,
154                                 tile_start_k,
155                                 mask_multicast_n,
156                                 &queue_full[queue_idx]
157                             );
```

```
158                         }
159                     } else {
160                         ptx_load_tile_tma_2d(
161                             &tiles_s.queue[queue_idx].a[0][0],
162                             &A_tm,
163                             tile_coord_m * T_m,
164                             tile_start_k,
165                             &queue_full[queue_idx]
166                         );
167                     }
168
169                     if constexpr (CLUSTER_DIM_m > 1) {
170                         if (cluster_lane_m == 0) {
171                             ptx_load_tile_tma_multicast_2d(
172                                 &tiles_s.queue[queue_idx].b[0][0],
173                                 &B_tm,
174                                 tile_coord_n * T_n,
175                                 tile_start_k,
176                                 mask_multicast_m,
177                                 &queue_full[queue_idx]
178                             );
179                         }
180                     } else {
181                         ptx_load_tile_tma_2d(
182                             &tiles_s.queue[queue_idx].b[0][0],
183                             &B_tm,
184                             tile_coord_n * T_n,
185                             tile_start_k,
186                             &queue_full[queue_idx]
187                         );
188                     }
189                 }
190             }
191         }
192     } else {
193         asm volatile("setmaxnreg.inc.sync.aligned.u32 %0;" :: "n"(240): "memory");
194         int queue_idx = 0;
195         int queue_phase = 0;
196         const int consumer_warpgroup_id =
197             (threadIdx.x - WARP_GROUP_SIZE) / WARP_GROUP_SIZE;
198         const int tile_start_m = consumer_warpgroup_id * WGMMA_m;
199         const int consumer_thread_id = threadIdx.x % WARP_GROUP_SIZE;
200         const uint thread_lane_idx_n = (consumer_thread_id % 32) % 4;
201
202         const int thread_store_offset_m = (
203             tile_start_m
204             + consumer_thread_id / 32 * 16
205             + (consumer_thread_id % 32) / 4
206         );
207         const int thread_store_offset_n =
208             ((consumer_thread_id % 32) % 4) * 2;
209
210         if (consumer_thread_id < CLUSTER_SIZE) {
211             for (int queue_idx = 0; queue_idx < QUEUE_SIZE; ++queue_idx) {
212                 ptx_arrive_barrier_across_cluster(
213                     &queue_empty[queue_idx],
214                     consumer_thread_id,
215                     1
216                 );
217             }
218         }
219
220         float C_accum[T_n/16][8];
221         for (int schedule_it = 0;
222              schedule_it < schedule_size_per_sm;
223              ++schedule_it) {
224             const int packed_tile = schedule_s[schedule_it];
225             if (packed_tile == -1) {
226                 break;
227             }
228
229             int tile_coord_m = packed_tile >> 16;
230             int tile_coord_n = packed_tile & 0xFFFF;
231
232             if constexpr (CLUSTER_DIM_n > 1) {
233                 tile_coord_n *= CLUSTER_DIM_n;
234                 tile_coord_n += cluster_lane_n;
235             }
236             if constexpr (CLUSTER_DIM_m > 1) {
237                 tile_coord_m *= CLUSTER_DIM_m;
238                 tile_coord_m += cluster_lane_m;
```

```
239                }
240
241            if (queue_idx == QUEUE_SIZE) {
242                queue_idx = 0;
243                queue_phase ^= 1;
244            }
245
246            ptx_wait_barrier(&queue_full[queue_idx], queue_phase);
247            asm volatile("wgmma.fence.sync.aligned;" ::: "memory");
248
249            wgmma<T_n, 0, 1, 1, 0, 0>(
250                C_accum,
251                &tiles_s.queue[queue_idx].a[tile_start_m][0],
252                &tiles_s.queue[queue_idx].b[0][0]
253            );
254
255            #pragma unroll
256            for (int wgmma_start_k = WGMMA_k;
257                 wgmma_start_k < T_k;
258                 wgmma_start_k += WGMMA_k) {
259                wgmma<T_n, 1, 1, 1, 0, 0>(
260                    C_accum,
261                    &tiles_s.queue[queue_idx].a[tile_start_m][wgmma_start_k],
262                    &tiles_s.queue[queue_idx].b[0][wgmma_start_k]
263                );
264            }
265
266            asm volatile("wgmma.commit_group.sync.aligned;" ::: "memory");
267            asm volatile("wgmma.wait_group.sync.aligned %0;" :: "n"(0): "memory");
268
269            if (consumer_thread_id < CLUSTER_SIZE) {
270                ptx_arrive_barrier_across_cluster(
271                    &queue_empty[queue_idx],
272                    consumer_thread_id,
273                    1
274                );
275            }
276
277            queue_idx++;
278
279            for (int tile_idx_k = 1;
280                 tile_idx_k < K / T_k;
281                 ++tile_idx_k, ++queue_idx) {
282                if (queue_idx == QUEUE_SIZE) {
283                    queue_idx = 0;
284                    queue_phase ^= 1;
285                }
286
287                ptx_wait_barrier(&queue_full[queue_idx], queue_phase);
288                asm volatile("wgmma.fence.sync.aligned;" ::: "memory");
289
290                #pragma unroll
291                for (int wgmma_start_k = 0;
292                     wgmma_start_k < T_k;
293                     wgmma_start_k += WGMMA_k) {
294                    wgmma<T_n, 1, 1, 1, 0, 0>(
295                        C_accum,
296                        &tiles_s.queue[queue_idx].a[tile_start_m][wgmma_start_k],
297                        &tiles_s.queue[queue_idx].b[0][wgmma_start_k]
298                    );
299                }
300
301                asm volatile("wgmma.commit_group.sync.aligned;" ::: "memory");
302                asm volatile("wgmma.wait_group.sync.aligned %0;" :: "n"(0): "memory");
303
304                if (consumer_thread_id < CLUSTER_SIZE) {
305                    ptx_arrive_barrier_across_cluster(
306                        &queue_empty[queue_idx],
307                        consumer_thread_id,
308                        1
309                    );
310                }
311            }
312
313            {
314                asm volatile("cp.async.bulk.wait_group.read 0;\n");
315                if (thread_lane_idx_n <= 1) {
316                    tiles_s.c_packed[
317                        thread_store_offset_m + 8 * thread_lane_idx_n
318                    ][0] = 0;
319                }
```

```
320                     __syncwarp();
321
322                 #pragma unroll
323                 for (int quadrant_slice_m = 0;
324                      quadrant_slice_m < 4;
325                      quadrant_slice_m += 2) {
326                     int quadrant_store_offset_m =
327                         thread_store_offset_m + quadrant_slice_m * 4;
328
329                     #pragma unroll
330                     for (int wgmma_slice_n = 0;
331                          wgmma_slice_n < T_n / 16;
332                          ++wgmma_slice_n) {
333                         int quadrant_store_offset_n =
334                             thread_store_offset_n + wgmma_slice_n * 16;
335
336                         #pragma unroll
337                         for (int quadrant_slice_n = 0;
338                              quadrant_slice_n < 8;
339                              quadrant_slice_n += 4) {
340                             #pragma unroll
341                             for (int element_n = 0;
342                                  element_n < 2;
343                                  ++element_n) {
344                                 if (
345                                     C_accum[wgmma_slice_n][
346                                         quadrant_slice_m
347                                         + quadrant_slice_n
348                                         + element_n
349                                     ] > 0
350                                 ) {
351                                     uint current_store_idx = __nv_atomic_fetch_add(
352                                         &tiles_s.c_packed[quadrant_store_offset_m][0],
353                                         1u,
354                                         __NV_ATOMIC_RELAXED,
355                                         __NV_THREAD_SCOPE_BLOCK
356                                     );
357
358                                     const uint32_t packed_value =
359                                         tile_coord_n * T_n
360                                         + quadrant_store_offset_n
361                                         + quadrant_slice_n * 2
362                                         + element_n
363                                         | (
364                                             static_cast<uint32_t>(
365                                                 __bfloat16_as_ushort(
366                                                     __float2bfloat16(
367                                                         C_accum[wgmma_slice_n][
368                                                             quadrant_slice_m
369                                                             + quadrant_slice_n
370                                                             + element_n
371                                                         ]
372                                                     )
373                                                 )
374                                             ) << 16
375                                         );
376
377                                     if constexpr (LOOP_OVERFLOW_STORAGE) {
378                                         tiles_s.c_packed[quadrant_store_offset_m][
379                                             (current_store_idx & (T_n_compressed - 1)) + 1
380                                         ] = packed_value;
381                                     } else {
382                                         tiles_s.c_packed[quadrant_store_offset_m][
383                                             current_store_idx + 1
384                                         ] = packed_value;
385                                     }
386                                 }
387                             }
388                         }
389                     }
390                 }
391
392                 asm volatile("fence.proxy.async.shared::cta;\n");
393                 asm volatile("bar.sync 10, 256;\n");
394
395                 if (threadIdx.x == 128) {
396                     ptx_store_transposed_tile_tma_3d<uint32_t, T_n_compressed>(
397                         &C_packed_tm,
398                         &tiles_s.c_packed[0][0],
399                         tile_coord_m * T_m,
400                         tile_coord_n * T_n_compressed
```

```
401                    );
402                    asm volatile("cp.async.bulk.commit_group;\n");
403                  }
404                }
405              }
406          }
407  }
```

*Listing 1.* Efficient matrix multiplication with TwELL output storage.

In Listing 1, we provide code listings with the device code for our kernel implementing a custom matmul with our new TwELL storage, which we use to run the gate projection in our model. We omit device functions wrapping longer PTX injections for readability. As explained in Section 3 in the main text, this kernel executes an efficient tiled matrix multiplication, loading the dense input and the dense weight matrix and storing the output values using the Tensor Memory Accelerator (TMA) introduced with Hopper GPUs, while storing the output in the TwELL format during the kernel's epilogue. The base kernel follows a persistent design with pipelined computation based on persistent cooperative kernels in CUTLASS (NVIDIA, 2025c) and open-source CUDA reproductions (Shankhdhar, 2024). Unlike CUTLASS, the tile scheduler follows a pre-constructed ordering based on a Hilbert curve to maximize the reuse of the L2 cache (Chatterjee et al., 1999; Shankhdhar, 2024). In practice, we opt to pack the TwELL values $h$, indices $h_I$, and number of non-zeros $h_{nz}$ in a single 32-bit matrix in $\mathbb{R}^{M \times N/C}$. This is done by placing the number of non-zeros for each tile row in

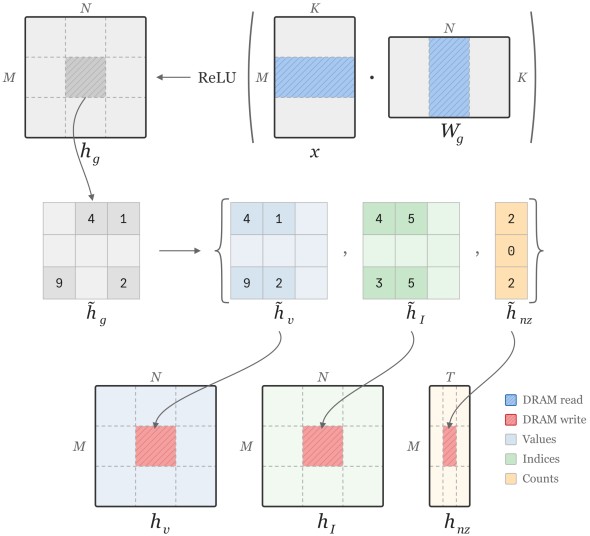

*Figure 8.* Illustration of the logic behind the efficient matrix multiplication with TwELL output storage.

the first column and fitting the 16-bit TwELL value and index in the remaining entries. This design ensures strong locality and allows storing and loading the number of non-zeros together with the first 31 TwELL indices and values in a single coalesced access. While this loses a storage position, we set TwELL compression factors very conservatively for each sparsity level, making the occurrence of overflow practically impossible. For instance, we set the compression factor to 8 for our recommended sparsity regularization studied in our main results, with models ranging from 39-24 average non-zeros and an expected chance of overflow of the order of $10^{-34}$. The TwELL conversion occurs when mapping the partial outputs of the asynchronous warpgroup level matmul instructions (WGMMA) from registers to shared memory via a fast CTA-scoped atomic operation with relaxed semantics. To avoid bank conflicts when resetting the number of non-zeros, we minimally pad the TwELL output with four extra elements in the last dimension. In this instance, we found our padding approach to work significantly better than swizzling, due to the lower register pressure introduced in the kernel's epilogue. In an alternative implementation, we also explored a different packing layout, placing the number of non-zeros across the diagonal of the first 32-dimensional subtile to cover all memory banks, an approach we found brought minimal throughput improvements at the cost of extra complexity. We note that for the non-gated variant of our models, we use this same kernel to perform the up projection, as this layer is the one determining the overall sparsity pattern of the tile.

```
1   template <
2       const int T_n,
3       const int T_n_compressed,
4       const int NUM_T_n,
5       const int OUT_DIM
6   >
7   __global__ __launch_bounds__(WARP_SIZE)
8   void mm_t2d_kernel(
9       const __nv_bfloat16* IN_d,
10      const uint32_t* GATE_OUT_twell_packed_d,
11      const __nv_bfloat16* UP_transposed_d,
12      const __nv_bfloat16* DOWN_d,
13      __nv_bfloat16* OUT_d
14  )
15  {
16      static_assert(
17          (OUT_DIM % STRIDE_8xWARP) == 0,
18          "OUT_DIM must be divisible by WARP_SIZE."
19      );
20      static_assert(T_n_compressed == WARP_SIZE,
21          "Warp-sync TwELL-to-dense assumes a 32-wide compressed tile.");
22
23      constexpr int NUM_LOAD_ITERS = OUT_DIM / STRIDE_8xWARP;
24      float OUT_accum[NUM_LOAD_ITERS][8] = {0.0f};
25      __nv_bfloat162 IN_cached[NUM_LOAD_ITERS][4];
26
27      IN_d += blockIdx.x * OUT_DIM + threadIdx.x * 8;
28      GATE_OUT_twell_packed_d += (
29          blockIdx.x * T_n_compressed * NUM_T_n + threadIdx.x
30      );
31      UP_transposed_d += threadIdx.x * 8;
32      DOWN_d += threadIdx.x * 8;
33      OUT_d += blockIdx.x * OUT_DIM + threadIdx.x * 8;
34
35      #pragma unroll
36      for (int iter_idx = 0; iter_idx < NUM_LOAD_ITERS; ++iter_idx) {
37          *reinterpret_cast<uint4*>(&IN_cached[iter_idx][0]) =
38              *reinterpret_cast<const uint4*>(
39                  IN_d + iter_idx * STRIDE_8xWARP
40              );
41      }
42
43      #pragma unroll 1
44      for (int tile_idx = 0; tile_idx < NUM_T_n; ++tile_idx) {
45          const int lane_tile_register =
46              GATE_OUT_twell_packed_d[tile_idx * T_n_compressed];
47          const int num_nonzeros =
48              __shfl_sync(0xFFFFFFFFu, lane_tile_register, 0);
49
50          #pragma unroll 1
51          for (int idx = 1; idx < num_nonzeros + 1; ++idx) {
52              const uint32_t compressed_idx_bf16 =
53                  __shfl_sync(0xFFFFFFFFu, lane_tile_register, idx);
54              const uint32_t nonzero_idx = compressed_idx_bf16 & 0xFFFFu;
55
56              float UP_OUT_accum = 0.0f;
57
58              #pragma unroll
59              for (int iter_idx = 0; iter_idx < NUM_LOAD_ITERS; ++iter_idx) {
60                  const uint4 packed_bfloats_x8 =
61                      *reinterpret_cast<const uint4*>(
62                          UP_transposed_d
63                          + nonzero_idx * OUT_DIM
64                          + iter_idx * STRIDE_8xWARP
65                      );
66                  const __nv_bfloat162 packed_bfloats_1 =
67                      *reinterpret_cast<const __nv_bfloat162*>(
68                          &packed_bfloats_x8.x
69                      );
70                  __nv_bfloat162 scaled_bfloats_1 =
71                      __hmul2(IN_cached[iter_idx][0], packed_bfloats_1);
72                  float2 scaled_floats_1 = __bfloat1622float2(scaled_bfloats_1);
73                  UP_OUT_accum += scaled_floats_1.x + scaled_floats_1.y;
74
75                  const __nv_bfloat162 packed_bfloats_2 =
76                      *reinterpret_cast<const __nv_bfloat162*>(
77                          &packed_bfloats_x8.y
78                      );
79                  __nv_bfloat162 scaled_bfloats_2 =
80                      __hmul2(IN_cached[iter_idx][1], packed_bfloats_2);
```

```
81              float2 scaled_floats_2 = __bfloat1622float2(scaled_bfloats_2);
82              UP_OUT_accum += scaled_floats_2.x + scaled_floats_2.y;
83
84              const __nv_bfloat162 packed_bfloats_3 =
85                  *reinterpret_cast<const __nv_bfloat162*>(
86                      &packed_bfloats_x8.z
87                  );
88              __nv_bfloat162 scaled_bfloats_3 =
89                  __hmul2(IN_cached[iter_idx][2], packed_bfloats_3);
90              float2 scaled_floats_3 = __bfloat1622float2(scaled_bfloats_3);
91              UP_OUT_accum += scaled_floats_3.x + scaled_floats_3.y;
92
93              const __nv_bfloat162 packed_bfloats_4 =
94                  *reinterpret_cast<const __nv_bfloat162*>(
95                      &packed_bfloats_x8.w
96                  );
97              __nv_bfloat162 scaled_bfloats_4 =
98                  __hmul2(IN_cached[iter_idx][3], packed_bfloats_4);
99              float2 scaled_floats_4 = __bfloat1622float2(scaled_bfloats_4);
100             UP_OUT_accum += scaled_floats_4.x + scaled_floats_4.y;
101         }
102
103         #pragma unroll
104         for (int butterfly_stride = WARP_SIZE / 2;
105              butterfly_stride > 0;
106              butterfly_stride /= 2) {
107             UP_OUT_accum += __shfl_xor_sync(
108                 0xFFFFFFFFu,
109                 UP_OUT_accum,
110                 butterfly_stride
111             );
112         }
113
114         const __nv_bfloat162 nonzero_feature =
115             __bfloat162bfloat162(
116                 __hmul(
117                     reinterpret_cast<const __nv_bfloat16*>(
118                         &compressed_idx_bf16
119                     )[1],
120                     __float2bfloat16_rn(UP_OUT_accum)
121                 )
122             );
123
124         #pragma unroll
125         for (int iter_idx = 0; iter_idx < NUM_LOAD_ITERS; ++iter_idx) {
126             const uint4 packed_bfloats_x8 =
127                 *reinterpret_cast<const uint4*>(
128                     DOWN_d
129                     + nonzero_idx * OUT_DIM
130                     + iter_idx * STRIDE_8xWARP
131                 );
132             const __nv_bfloat162 packed_bfloats_1 =
133                 *reinterpret_cast<const __nv_bfloat162*>(
134                     &packed_bfloats_x8.x
135                 );
136             __nv_bfloat162 scaled_bfloats_1 =
137                 __hmul2(nonzero_feature, packed_bfloats_1);
138             float2 scaled_floats_1 = __bfloat1622float2(scaled_bfloats_1);
139             OUT_accum[iter_idx][0] += scaled_floats_1.x;
140             OUT_accum[iter_idx][1] += scaled_floats_1.y;
141
142             const __nv_bfloat162 packed_bfloats_2 =
143                 *reinterpret_cast<const __nv_bfloat162*>(
144                     &packed_bfloats_x8.y
145                 );
146             __nv_bfloat162 scaled_bfloats_2 =
147                 __hmul2(nonzero_feature, packed_bfloats_2);
148             float2 scaled_floats_2 = __bfloat1622float2(scaled_bfloats_2);
149             OUT_accum[iter_idx][2] += scaled_floats_2.x;
150             OUT_accum[iter_idx][3] += scaled_floats_2.y;
151
152             const __nv_bfloat162 packed_bfloats_3 =
153                 *reinterpret_cast<const __nv_bfloat162*>(
154                     &packed_bfloats_x8.z
155                 );
156             __nv_bfloat162 scaled_bfloats_3 =
157                 __hmul2(nonzero_feature, packed_bfloats_3);
158             float2 scaled_floats_3 = __bfloat1622float2(scaled_bfloats_3);
159             OUT_accum[iter_idx][4] += scaled_floats_3.x;
160             OUT_accum[iter_idx][5] += scaled_floats_3.y;
161
```

```
162            const __nv_bfloat162 packed_bfloats_4 =
163                *reinterpret_cast<const __nv_bfloat162*>(
164                    &packed_bfloats_x8.w
165                );
166            __nv_bfloat162 scaled_bfloats_4 =
167                __hmul2(nonzero_feature, packed_bfloats_4);
168            float2 scaled_floats_4 = __bfloat1622float2(scaled_bfloats_4);
169            OUT_accum[iter_idx][6] += scaled_floats_4.x;
170            OUT_accum[iter_idx][7] += scaled_floats_4.y;
171          }
172        }
173     }
174
175     #pragma unroll
176     for (int iter_idx = 0; iter_idx < NUM_LOAD_ITERS; ++iter_idx) {
177         __nv_bfloat162 __align__(8) packed_bfloats_x8[4];
178         packed_bfloats_x8[0] = __floats2bfloat162_rn(
179             OUT_accum[iter_idx][0], OUT_accum[iter_idx][1]
180         );
181         packed_bfloats_x8[1] = __floats2bfloat162_rn(
182             OUT_accum[iter_idx][2], OUT_accum[iter_idx][3]
183         );
184         packed_bfloats_x8[2] = __floats2bfloat162_rn(
185             OUT_accum[iter_idx][4], OUT_accum[iter_idx][5]
186         );
187         packed_bfloats_x8[3] = __floats2bfloat162_rn(
188             OUT_accum[iter_idx][6], OUT_accum[iter_idx][7]
189         );
190
191         *reinterpret_cast<uint4*>(OUT_d + iter_idx * STRIDE_8xWARP) =
192             *reinterpret_cast<uint4*>(packed_bfloats_x8);
193     }
194 }
```

*Listing 2.* Fused up and downprojection leveraging gate projections in TwELL format.

In Listing 2, we provide code listings with the device code for our kernel implementing the custom fused up and down projection kernel that leverages the gate projections stored in the TwELL format. As explained in Section 3 in the main text, this kernel is launched on a grid of warp-sized CTAs and fuses the two operations by keeping in memory the input dense feature row and an accumulator. Then, iterating first statically through the TwELL tiles and then dynamically through the number of non-zeros in each tile, it loads the corresponding gate index, which directly maps to a unique column of the up projection and row of the down projection weight matrices. The kernel computes the up-projected feature from a dot product between the input dense feature row and the up projection weight column, multiplies it by the gate value, and finally uses it to scale the down projection weight row before accumulating the output. To ensure coalesced access, we note that the up projection weight matrix is stored in transposed format. This version of the kernel is specialized to handle the case where $T_n = 256$ and the compression ratio is 8, leading to a total of 32

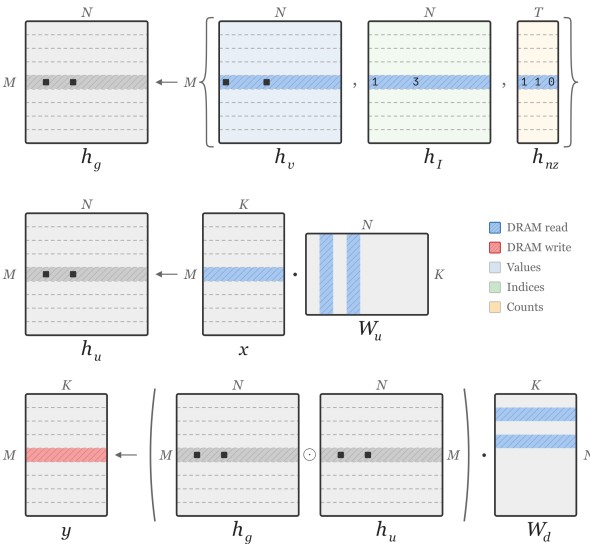

*Figure 9.* Illustration of the fused up and downprojection leveraging gate projections in TwELL format.

elements for each packed TwELL tile. In this specific instance, we load both the number of non-zeros and all the indices and values for the tile in a single fully coalesced access over the CTA's warp, which later allows loading the full TwELL tile information via minimal warp register shuffle operations without incurring any shared memory overheads. In preliminary experiments, we also found that re-ordering the kernel calls in descending order of non-zeros can further accelerate performance with low batch sizes. However, we note that we did not find this optimization necessary with large batches and omitted it for simplicity.

```
1  template <
2      const int T_n,
3      const int T_n_compressed,
4      const int NUM_T_n,
5      const int OUT_DIM,
6      const int SPLIT_OUT_DIM
7  >
8  __global__ __launch_bounds__(32)
9  void mm_t2d_kernel(
10     const uint32_t* IN_twell_packed_d,
11     const __nv_bfloat16* DOWN_d,
12     __nv_bfloat16* OUT_d
13 )
14 {
15     static_assert(
16         (SPLIT_OUT_DIM % STRIDE_8xWARP) == 0,
17         "OUT_DIM must be divisible by WARP_SIZE."
18     );
19     static_assert(
20         (OUT_DIM % SPLIT_OUT_DIM) == 0,
21         "OUT_DIM must be divisible by SPLIT_OUT_DIM."
22     );
23     static_assert(T_n_compressed == WARP_SIZE,
24         "Warp-sync TwELL-to-dense assumes a 32-wide compressed tile.");
25
26     float OUT_accum[OUT_DIM / STRIDE_8xWARP][8] = {0.0f};
27     constexpr int NUM_LOAD_ITERS = SPLIT_OUT_DIM / STRIDE_8xWARP;
28
29     IN_twell_packed_d += blockIdx.x * T_n_compressed * NUM_T_n + threadIdx.x;
30     DOWN_d += threadIdx.x * 8 + blockIdx.y * SPLIT_OUT_DIM;
31     OUT_d += blockIdx.x * OUT_DIM + threadIdx.x * 8 + blockIdx.y * SPLIT_OUT_DIM;
32
33     #pragma unroll 1
34     for (int tile_idx = 0; tile_idx < NUM_T_n; ++tile_idx) {
35         const int lane_tile_register =
36             IN_twell_packed_d[tile_idx * T_n_compressed];
37         const int num_nonzeros =
38             __shfl_sync(0xFFFFFFFFu, lane_tile_register, 0);
39
40         #pragma unroll 1
41         for (int idx = 1; idx < num_nonzeros + 1; ++idx) {
42             const uint32_t compressed_idx_bf16 =
43                 __shfl_sync(0xFFFFFFFFu, lane_tile_register, idx);
44             const uint32_t nonzero_idx = compressed_idx_bf16 & 0xFFFFu;
45             const __nv_bfloat162 nonzero_feature =
46                 __bfloat162bfloat162(
47                     reinterpret_cast<const __nv_bfloat16*>(
48                         &compressed_idx_bf16
49                     )[1]
50                 );
51
52             #pragma unroll
53             for (int iter_idx = 0; iter_idx < NUM_LOAD_ITERS; ++iter_idx) {
54                 const uint4 packed_bfloats_x8 =
55                     *reinterpret_cast<const uint4*>(
56                         DOWN_d
57                         + nonzero_idx * OUT_DIM
58                         + iter_idx * STRIDE_8xWARP
59                     );
60                 const __nv_bfloat162 packed_bfloats_1 =
61                     *reinterpret_cast<const __nv_bfloat162*>(
62                         &packed_bfloats_x8.x
63                     );
64                 __nv_bfloat162 scaled_bfloats_1 =
65                     __hmul2(nonzero_feature, packed_bfloats_1);
66                 float2 scaled_floats_1 = __bfloat1622float2(scaled_bfloats_1);
67                 OUT_accum[iter_idx][0] += scaled_floats_1.x;
68                 OUT_accum[iter_idx][1] += scaled_floats_1.y;
69
70                 const __nv_bfloat162 packed_bfloats_2 =
71                     *reinterpret_cast<const __nv_bfloat162*>(
72                         &packed_bfloats_x8.y
73                     );
74                 __nv_bfloat162 scaled_bfloats_2 =
75                     __hmul2(nonzero_feature, packed_bfloats_2);
76                 float2 scaled_floats_2 = __bfloat1622float2(scaled_bfloats_2);
77                 OUT_accum[iter_idx][2] += scaled_floats_2.x;
78                 OUT_accum[iter_idx][3] += scaled_floats_2.y;
79
80                 const __nv_bfloat162 packed_bfloats_3 =
```

```
 81                    *reinterpret_cast<const __nv_bfloat162*>(
 82                        &packed_bfloats_x8.z
 83                    );
 84                __nv_bfloat162 scaled_bfloats_3 =
 85                    __hmul2(nonzero_feature, packed_bfloats_3);
 86                float2 scaled_floats_3 = __bfloat1622float2(scaled_bfloats_3);
 87                OUT_accum[iter_idx][4] += scaled_floats_3.x;
 88                OUT_accum[iter_idx][5] += scaled_floats_3.y;
 89
 90                const __nv_bfloat162 packed_bfloats_4 =
 91                    *reinterpret_cast<const __nv_bfloat162*>(
 92                        &packed_bfloats_x8.w
 93                    );
 94                __nv_bfloat162 scaled_bfloats_4 =
 95                    __hmul2(nonzero_feature, packed_bfloats_4);
 96                float2 scaled_floats_4 = __bfloat1622float2(scaled_bfloats_4);
 97                OUT_accum[iter_idx][6] += scaled_floats_4.x;
 98                OUT_accum[iter_idx][7] += scaled_floats_4.y;
 99            }
100        }
101    }
102
103    #pragma unroll
104    for (int iter_idx = 0; iter_idx < NUM_LOAD_ITERS; ++iter_idx) {
105        __nv_bfloat162 __align__(8) packed_bfloats_x8[4];
106        packed_bfloats_x8[0] = __floats2bfloat162_rn(
107            OUT_accum[iter_idx][0], OUT_accum[iter_idx][1]
108        );
109        packed_bfloats_x8[1] = __floats2bfloat162_rn(
110            OUT_accum[iter_idx][2], OUT_accum[iter_idx][3]
111        );
112        packed_bfloats_x8[2] = __floats2bfloat162_rn(
113            OUT_accum[iter_idx][4], OUT_accum[iter_idx][5]
114        );
115        packed_bfloats_x8[3] = __floats2bfloat162_rn(
116            OUT_accum[iter_idx][6], OUT_accum[iter_idx][7]
117        );
118
119        *reinterpret_cast<uint4*>(OUT_d + iter_idx * STRIDE_8xWARP) =
120            *reinterpret_cast<uint4*>(packed_bfloats_x8);
121    }
122 }
```

*Listing 3.* Down projection leveraging up projections from the TwELL format for the non-gated model variants.

As mentioned in Section 2 of the main text, together with modern gated LLMs, we also provide specific kernels that support older non-gated variants, which we empirically evaluate in Appendix C. In Listing 3, we provide code listings with the device code for our kernel implementing the custom down projection kernel that leverages up projection activations stored in the TwELL format for these experiments. Similarly to the fused kernel explained in Section 3 and examined above, this kernel is launched on a grid of warp-sized CTAs and reads the sparsity pattern, this time from the up projection activations stored in the TwELL format. This time, the kernel maintains in memory the out projection and a float32 accumulator for a small output segment. Then, it first statically iterates through the TwELL tile and then dynamically iterates over the number of non-zeros in each tile. At each iteration, it loads the non-zero index and the corresponding activation down projection column segment, before multiplying the two and accumulating the result. In contrast to our fused kernel, where we have to consider full rows on the input and output to perform dot products between the input features and the up projection weights, introducing the split formulation in this kernel is a deliberate and purposeful choice: by introducing trivial duplication of the non-zero reads we can further increase parallelism, reduce register pressure, increase occupancy, and hide longer latencies from uneven sparsity. In practice, we note that using a split dimension of half the base output dimension, leading to two CTAs per output row, appears optimal on our Hopper GPUs.

## A.2. Training Kernels Selection

```
1   __global__ void twell_to_ell_kernel(
2       const __nv_bfloat16* __restrict__ C_vals,
3       const uint8_t* __restrict__ C_idx,
4       const uint32_t* __restrict__ C_nnz,
5       __nv_bfloat16* __restrict__ ell_val,
6       int16_t* __restrict__ ell_col,
7       int32_t* __restrict__ row_nnz,
8       float* __restrict__ l0_out,
9       float* __restrict__ l1_out,
10      int M,
11      int N_TILES,
12      int BW,
13      int ELL_W,
14      int T_n
15  )
16  {
17      const int row = blockIdx.x * blockDim.y + threadIdx.y;
18      if (row >= M) {
19          return;
20      }
21
22      const int tid = threadIdx.x;
23      int cnt = (tid < N_TILES) ? C_nnz[(size_t)tid * M + row] : 0;
24      cnt = min(cnt, BW);
25
26      int offset = cnt;
27      for (int delta = 1; delta < WARP_SIZE; delta <<= 1) {
28          const int recv = __shfl_up_sync(0xFFFFFFFFu, offset, delta);
29          if (tid >= delta) {
30              offset += recv;
31          }
32      }
33
34      const int start = offset - cnt;
35      const int total =
36          __shfl_sync(0xFFFFFFFFu, offset, min(N_TILES - 1, WARP_SIZE - 1));
37
38      const __nv_bfloat16* sv =
39          C_vals + (size_t)row * N_TILES * BW + (size_t)tid * BW;
40      const uint8_t* si =
41          C_idx + (size_t)row * N_TILES * BW + (size_t)tid * BW;
42
43      float l0_acc = 0.0f;
44      float l1_acc = 0.0f;
45      if (l0_out) {
46          const float inv_M = 1.0f / (float)M;
47          l0_acc = (float)cnt * inv_M;
48          for (int i = 0; i < cnt; ++i) {
49              l1_acc += __bfloat162float(sv[i]) * inv_M;
50          }
51      }
52
53      if (cnt > 0 && start < ELL_W) {
54          const int copy_n = min(cnt, ELL_W - start);
55          __nv_bfloat16* dv = ell_val + (size_t)row * ELL_W + start;
56          int16_t* dc = ell_col + (size_t)row * ELL_W + start;
57          for (int i = 0; i < copy_n; ++i) {
58              dv[i] = sv[i];
59              dc[i] = (int16_t)(si[i]) + (int16_t)(tid * T_n);
60          }
61      }
62
63      if (tid == 0) {
64          row_nnz[row] = total;
65      }
66
67      if (l0_out) {
68          for (int s = 16; s > 0; s >>= 1) {
69              l0_acc += __shfl_down_sync(0xFFFFFFFFu, l0_acc, s);
70              l1_acc += __shfl_down_sync(0xFFFFFFFFu, l1_acc, s);
71          }
72          if (tid == 0) {
73              atomicAdd(l0_out, l0_acc);
74              if (l1_out) {
75                  atomicAdd(l1_out, l1_acc);
76              }
77          }
78      }
79  }
```

*Listing 4.* Conversion from TwELL to the hybrid format logic.

In Listing 4, we provide code listings with the device code for our training kernel used to convert gate activations stored in the TwELL format into the compact ELL component of our hybrid training representation while accumulating $L_0$ and $L_1$ statistics. As discussed in Section 3, the conversion dynamically partitions the rows based on the non-zero counts. We allocate a warp to each row, and let each thread read the number of active entries in a single tile. We then use warp register shuffles to obtain an inclusive prefix scan and determine the starting offset of that tile within the destination ELL row. This design allows for directly compacting the tiled representation into contiguous row-wise ELL storage without requiring any synchronization beyond warp-level or shared memory accesses. The kernel writes the true row occupancy to $row\_nnz$ even when the row exceeds the configured ELL width $ELL\_W$, allowing overflow rows to be detected and promoted to the dense tail of the hybrid format. During training, each warp also reduces simple $L_0$ and $L_1$ sparsity statistics to compute the sparsity levels and L1 loss before issuing a single atomic update.

```
1  __global__ void matmul_save_sparse_like_ell(
2      bfloat16* A,
3      bfloat16* B_T,
4      ELL* out,
5      int M,
6      int K,
7      int N
8  )
9  {
10     const int row = blockIdx.x;
11     const int ell_n = out->row_counts[row];
12     if (ell_n == 0 || ell_n > ELL_WIDTH) {
13         return;
14     }
15
16     bfloat16* A_row_ptr = A + row * K;
17     const int lane_id = threadIdx.x & 31;
18     const int warp_id = threadIdx.x >> 5;
19     const int num_warps = blockDim.x >> 5;
20     const int num_chunks = K / 8;
21
22     for (int out_idx = warp_id; out_idx < ell_n; out_idx += num_warps) {
23         const int col = out->cols[row * ELL_WIDTH + out_idx];
24         bfloat16* B_row_ptr = B_T + col * K;
25         float acc = 0.0f;
26
27         for (int chunk_base = 0; chunk_base < num_chunks; chunk_base += 32) {
28             const int chunk = chunk_base + lane_id;
29             if (chunk >= num_chunks) {
30                 break;
31             }
32
33             int4 a_raw = *(int4*)(A_row_ptr + chunk * 8);
34             int4 b_raw = *(int4*)(B_row_ptr + chunk * 8);
35             bfloat16_2* a_vec = (bfloat16_2*)&a_raw;
36             bfloat16_2* b_vec = (bfloat16_2*)&b_raw;
37
38             for (int t = 0; t < 4; ++t) {
39                 float2 af = bfloat1622float2(a_vec[t]);
40                 float2 bf = bfloat1622float2(b_vec[t]);
41                 acc = fmaf(af.x, bf.x, acc);
42                 acc = fmaf(af.y, bf.y, acc);
43             }
44         }
45
46         for (int offset = 16; offset > 0; offset >>= 1) {
47             acc += __shfl_xor_sync(0xFFFFFFFFu, acc, offset);
48         }
49         if (lane_id == 0) {
50             out->vals[row * ELL_WIDTH + out_idx] = float2bfloat16(acc);
51         }
52     }
53 }
```

*Listing 5.* Dense-to-hybrid matmul for populating the sparse ELL component during training using CUDA cores.

In Listing 5, we provide code listings of our custom kernel used to perform the efficient dense-to-hybrid matmuls used during training, focusing on the sparse component. This kernel shows the logic of the dynamic hybrid partitioning, skipping the sparse operation in the non-zeros is recognized to exceed the size of the aggressively compact ELL format. The kernel takes two dense matrices, $A$ and $B$ (provided as $B_T$), and a pre-allocated ELL output "out" of shape $M \times N$, whose column indices encode the sparsity pattern to be evaluated. Rather than computing all $MN$ outputs, each thread block processes a single output row and iterates only over the column indices stored for that row in out. For each selected column, the kernel computes the dot product between $A[row, :]$ and $B_T[col, :]$. To maximize coalescing and enable vectorized memory accesses, $B$ is stored transposed so that rows of $B_T$ are contiguous in memory. To maximize throughput with bfloat16, threads load $A$ and $B_T$ in 128-bit transactions (8 bfloat16 values at a time) and accumulate in float32 using fused multiply-adds. Each warp reduces partial sums using shuffle-based reduction, and the final value is written to the corresponding slot in the ELL value array. This design aligns with ELL's row-oriented storage: the sparsity pattern is known up front, so the kernel avoids both dense materialization and irregular gathers beyond the indexed rows of $B_T$. In the forward pass, we use this kernel to compute only the entries of the up projection operation $xW_u$ that will survive the subsequent gating, by copying the sparsity pattern from $h_g$ into out and filling its values with the corresponding dot products. In the backward pass, the same kernel is reused for masked gradient matmuls that share a known sparsity pattern. For instance, we

use it to compute $\nabla h = \nabla y, W_d^T$. Rows that exceed the ELL capacity are handled by routing the overflow to the dense backup matrix and computing that portion with Tensor Core–optimized kernels, as described in Algorithm 3, and they are multiplied by a binary mask containing the sparsity pattern to be applied.

```
1  __global__ void hybrid_to_dense_mamtul(
2      ELL* A,
3      bfloat16* B,
4      bfloat16* C,
5      int M,
6      int N,
7      int K
8  )
9  {
10     const int row = blockIdx.x;
11     const int ell_n = A->row_counts[row];
12     if (ell_n == 0 || ell_n > ELL_WIDTH) {
13         return;
14     }
15
16     bfloat16* A_row_vals = A->vals + row * ELL_WIDTH;
17     uint16_t* A_row_idxs = A->cols + row * ELL_WIDTH;
18
19     for (int n_out = threadIdx.x * 8; n_out < N; n_out += 8 * blockDim.x) {
20         float2 acc[4];
21         for (int i = 0; i < 4; ++i) {
22             acc[i] = make_float2(0.f, 0.f);
23         }
24
25         for (int k = 0; k < ELL_WIDTH; ++k) {
26             if (k >= ell_n) {
27                 break;
28             }
29
30             const bfloat16 a_val = A_row_vals[k];
31             const uint16_t col_idx = A_row_idxs[k];
32             bfloat16* B_row_ptr = B + col_idx * N + n_out;
33             int4 b_vec_raw = *(int4*)(B_row_ptr);
34             bfloat162* b_vec = (bfloat162*)(&b_vec_raw);
35             const float a = bfloat162float(a_val);
36
37             for (int t = 0; t < 4; ++t) {
38                 float2 b_f32 = bfloat1622float2(b_vec[t]);
39                 acc[t].x = fmaf(a, b_f32.x, acc[t].x);
40                 acc[t].y = fmaf(a, b_f32.y, acc[t].y);
41             }
42         }
43
44         bfloat162* C_ptr = (bfloat162*)(C + row * N + n_out);
45         for (int i = 0; i < 4; ++i) {
46             C_ptr[i] = float22bfloat162(acc[i]);
47         }
48     }
49 }
```

*Listing 6.* Hybrid-to-dense sparse matmul using the ELL component during training using CUDA cores.

In Listing 6, we provide code listings of our custom kernel used to perform the efficient hybrid-to-dense used during training, focusing on the sparse component. Again, this kernel implements the same dynamic hybrid partitioning logic, skipping the sparse operation in the non-zeros is recognized to exceed the size of the aggressively compact ELL format. In particular, the kernel computes a sparse–dense matrix multiplication $C = AB$, where $A$ is stored in ELL format and $B$ and $C$ are dense row-major matrices. The kernel maps one CTA per output row of $C$, which aligns naturally with ELL's row-wise storage and lets the CTA reuse the same sparse row metadata while sweeping across the output columns. Within a CTA, threads partition the output row into contiguous column tiles. For each tile, they iterate over the non-zeros in the corresponding ELL row of $A$ and accumulate contributions of the form $a \cdot B[col_idx, :]$ into $C[row, :]$. To maximize memory throughput for bfloat16, the kernel accesses $B$ using 128-bit SIMD loads, so that each thread processes 8 output elements at a time. Accumulation is performed in float32, and the results are written back in vectorized form, providing a simple and efficient SpMM for the fixed-width ELL layout. Rows that exceed the ELL capacity are handled by routing the overflow to the dense backup matrix and computing that portion with Tensor Core–optimized kernels, as described in Algorithm 3. This kernel is used in the forward pass to compute the feedforward layer output. In the backward pass, it is also used to compute gradients with respect to the layer parameters as well as the input activations.

```
1   __global__ void hybrid_transpose(
2       ELL* A,
3       ELL* A_T,
4       bfloat16* tail_A,
5       bfloat16_t* tail_A_T,
6       int M,
7       int N
8   )
9   {
10      for (int row = blockIdx.x; row < M; row += gridDim.x) {
11          const int nnz_row = A->row_counts[row];
12          if (nnz_row == 0 || nnz_row > ELL_WIDTH) {
13              continue;
14          }
15
16          for (int k = threadIdx.x; k < nnz_row; k += blockDim.x) {
17              const uint16_t col = A->cols[row * ELL_WIDTH + k];
18              const bfloat16 val = A->vals[row * ELL_WIDTH + k];
19              const int out_row = col;
20              const int out_col = row;
21              const int pos = atomicAdd(A_T->row_counts[out_row], 1);
22
23              if (pos < ELL_WIDTH) {
24                  const size_t addr = out_row * ELL_WIDTH + pos;
25                  A_T->cols[addr] = out_col;
26                  A_T->vals[addr] = val;
27              } else {
28                  const int d_row =
29                      get_or_allocate_dense_row(out_row, A_T->tail_map);
30                  tail_A_T[d_row * M + out_col] = val;
31              }
32          }
33      }
34
35      for (int d_row = blockIdx.x; d_row < A->tail_rows; d_row += gridDim.x) {
36          const int src_row = A->tail_map_reverse[d_row];
37          bfloat16_t* src = tail_A + d_row * N;
38
39          for (int col0 = threadIdx.x * 8; col0 < N; col0 += blockDim.x * 8) {
40              int4 raw = *(int4*)(src + col0);
41              if ((raw.x | raw.y | raw.z | raw.w) == 0) {
42                  continue;
43              }
44
45              for (int i = 0; i < 8; ++i) {
46                  const bfloat16_t val = unpack_element(&raw, i);
47                  if (val == 0.0f) {
48                      continue;
49                  }
50
51                  const int out_row = col0 + i;
52                  const int out_col = src_row;
53                  const int pos = atomicAdd(A_T->row_counts[out_row], 1);
54
55                  if (pos < ELL_WIDTH) {
56                      const size_t addr = out_row * ELL_WIDTH + pos;
57                      A_T->cols[addr] = out_col;
58                      A_T->vals[addr] = val;
59                  } else {
60                      const int dense_row =
61                          get_or_allocate_dense_row(out_row, A_T->tail_map);
62                      tail_A_T[dense_row * M + out_col] = val;
63                  }
64              }
65          }
66      }
67  }
```

*Listing 7.* Transposition of the hybrid sparse used during training.

In Listing 7, we provide code listings of our custom kernel used to perform efficient transposition of a matrix stored in our hybrid training format. The kernel takes as input a matrix $A$ and produces $A_T$ in the same representation: an ELL matrix, plus a dense backup for rows that overflow the maximum number of non-zeros. It operates in two parts. First, it transposes the non-overflow rows stored in the ELL structure by iterating over each row's non-zeros and inserting them into the corresponding row of $A_T$ (since a non-zero at (row, col) becomes an entry in row col of the transpose). Because many source rows may map to the same destination row, the kernel uses atomic increments to reserve an insertion slot. If

the destination row still has capacity, the entry is written into the ELL arrays of $A_T$; otherwise, it is routed to the dense backup, using a per-row mapping that allocates a dense-tail row on demand. Second, it handles the overflow rows that are materialized in the dense tail of $A$. These rows are scanned in vectorized chunks (128-bit loads, i.e., 8 bfloat16 values at a time) with a fast zero check to skip all-zero vectors. Only non-zero elements are emitted into $A_T$ using the same hybrid partitioning logic. This approach keeps transposition efficient while preserving the hybrid format and avoiding expensive conversions to more general sparse layouts. After this kernel completes, we launch a small helper kernel to copy the entries stored in the ELL arrays for rows that overflowed into the corresponding dense-backup rows. We note that the necessity of this final small step comes from the fact that dense rows are only allocated and populated after the ELL slots for a given output row have been exhausted.

# B. Hyperparameters and Datasets

## B.1. Training Details

*Table 2.* Default Hyperparameters for Pretraining Sparse and Non-Sparse LLMs.

| Hyperparameter | Gated LLM | Non-Gated LLM |
|---|---|---|
| **Model architecture** | | |
| Hidden size | 2048 | 2048 |
| Hidden MLP size (intermediate) | 5632 | 8192 |
| Gated MLP | true | false |
| Hidden activation | ReLU | ReLU |
| Number of hidden layers | 8/18/28/38 | 8/18/28/38 |
| Number of attention heads | 32 | 32 |
| Number of key–value heads | 32 | 32 |
| Head dimension | 64 | 64 |
| Attention bias | false | false |
| Attention dropout | 0.0 | 0.0 |
| Initializer range | 0.02 | 0.02 |
| RoPE $\theta$ | 10,000 | 10,000 |
| Tied word embeddings | true | true |
| Vocabulary size | 49,152 | 49,152 |
| Tokenizer | GPT2 | GPT2 |
| Computation dtype | bfloat16 | bfloat16 |
| MLP bias | false | false |
| **Training setup** | | |
| Dataset | fineweb | finewebB |
| Maximum sequence length | 2048 | 2048 |
| Tokens per training step | 1,048,576 | 1,048,576 |
| Training steps | 10K/20K/30K/40K | 10K/20K/30K/40K |
| Total training tokens | 10.49B/20.97B/31.46B/41.94B | 10.49B/20.97B/31.46B/41.94B |
| **Optimization** | | |
| Optimizer | AdamW | AdamW |
| Learning rate | $1.0 \times 10^{-3}$ | $1.0 \times 10^{-3}$ |
| Weight decay | 0.1 | 0.1 |
| Adam parameters $(\beta_1, \beta_2, \epsilon)$ | $(0.9, 0.95, 1\times10^{-8})$ | $(0.9, 0.95, 1\times10^{-8})$ |
| Learning rate scheduler | Cosine decay | Cosine decay |
| Warmup steps | 600 | 600 |
| Max grad norm | 1.0 | 1.0 |

As explained in Section 4 of the main text, our sparse models and dense baselines in the main text implement a "Transformer++" architecture with gated feedforward blocks, as common in recent LLMs such as Qwen and Llama (Touvron et al., 2023; Yang et al., 2024). Moreover, in Appendix C, we also collect results on a non-gated variant of the same architecture, more similar to the traditional architecture, more similar to the original transformer design (Vaswani et al., 2017). We train all models using the fineweb (Penedo et al., 2024). In particular, we consider a deduplicated version of the fineweb-edu split, obtained by from an open corpus used to pretrain the SmolLM family of models (Allal et al., 2024). We note that all our models are trained with the chinchilla-optimal number of tokens (Hoffmann et al., 2022): around 10B tokens for our 0.5B models, 20B tokens for our 1B models, 30B tokens for our 1.5B models, and 40B tokens for our 2B models.

We provide a full list of hyperparameters and training specifications for our training settings and models in Table 2. For all models, we use context lengths of 2048 tokens, with batches of 512 sequences, resulting in a global batch size of 1M tokens. For our gated variant, we use a dimensionality of 2048 and a hidden dimension of 5632 in the feedforward blocks, roughly eight-thirds of the hidden size. The main difference with the non-gated variant is that we use a much larger intermediate size of 8192, four times the hidden size, leading to the same total number of parameters. We note that both these choices are considered optimal in the current literature with larger model design practices. When varying model sizes, we modify the number of layers to achieve the target parameter counts. In practice, modern small models have also considered shifting even more of the parameters and flops to the feedforward blocks: for instance, the 1B model of the llama 3 family has a

*Table 3.* Comparison of performance and efficiency statistics of sparse LLMs leveraging our kernels with traditional gated models using both ReLU and SiLU activations (Hendrycks, 2016; Ramachandran et al., 2017; Shazeer, 2020).

| Model scale | Activation | Sparse | L1 coeff. | Mean task accuracy | Cross-entropy | # non-zeros | Forward execution (input tokens/ms) | | Energy per token (mJ) | |
|---|---|---|---|---|---|---|---|---|---|---|
| 1.5B params 30B tokens | ReLU | ✗ | 0 | 46.4% | 2.255 | 911 | 117.1 | (0.0%) | 5.77 | (0.0%) |
| | SiLU | ✗ | 0 | 47.1% | 2.240 | 5632 | 116.5 | (-0.5%) | 5.82 | (+0.1%) |
| | ReLU | ✓ | $2 \times 10^{-5}$ | 46.2% | 2.297 | 29 | 138.0 | (+17.9%) | 5.07 | (-12.1%) |

feedforward size of 4x the hidden size even while implementing the gated design (Grattafiori et al., 2024). While the gains from our sparse kernels could be even greater in these settings, we opted for a more conservative design to avoid biasing our results toward smaller models.

To optimize our models, we use the AdamW optimizer (Loshchilov & Hutter, 2017) with a weight decay of 0.1 and a cosine learning rate schedule starting from a peak learning rate of $1.0 \times 10^{-3}$, after a small warmup of 600 steps. We use the default Adam parameters of $(\beta_1, \beta_2, \epsilon) = (0.9, 0.95, 1 \times 10^{-8})$ and clip gradients at a maximum norm of 1.0. Our vocabulary of tokens comes from a GPT2 tokenizer (Radford et al., 2019). We train using standard mixed precision with the bfloat16 format, with our optimizer states stored in full precision.

## B.2. Task Evaluation Details

We evaluate our models using cloze-formulation scores on seven standard downstream multiple-choice benchmarks that probe logical and commonsense reasoning after pretraining. In particular, we consider ARC (Easy and Hard versions) (Clark et al., 2018), a grade-school science question answering benchmark comprising both Easy and Challenge subsets, with the latter designed to defeat simple retrieval- and co-occurrence-based baselines; HellaSwag (Zellers et al., 2019), a commonsense sentence completion task that was designed for counterintuitive LLM challenge; OpenBookQA (Mihaylov et al., 2018), focused on probing curated sets of science-based and commonsense knowledge; PIQA (Bisk et al., 2020), a benchmark benchmark focused on physical commonsense reasoning; WinoGrande (Sakaguchi et al., 2021), a Winograd-style large-scale conference benchmark; and CommonsenseQA (Talmor et al., 2019), evaluating broader commonsense reasoning. We follow standard evaluation protocols and hyperparameters for formatting the input questions.

### B.2.1. SPARSE DATA STRUCTURES SIZING

We note that the hybrid training format proposed in this paper introduces two core hyperparameters necessary to fulfill its targeted static allocation design: the ELL maximum number of elements per row, and the number of rows in the dense matrix that stores overflowing elements. Both hyperparameters effectively control a trade-off between performance and memory savings, making their value partially dependent on the sparsity level. Moreover, because sparsity can change abruptly during training, we evaluate a set of sizes that can tolerate sudden decreases in sparsity while remaining performant. In practice, we find that setting the maximum number of elements to 128, and the maximum number of backup rows to one-eighth of the token batch size, to be a robust choice for all sparsity levels above $1.5 \times 10^{-5}$. Moreover, below this point, simply doubling the ELL non-zeros prevents other instabilities. However, we note that with prior knowledge of the sparsity evolution, these structures can often be made smaller within training itself. For example, for $L_1 = 1 \times 10^{-4}$, we observe that after training stabilizes, we can reduce the number of rows in the dense overflow matrix to 512, enabling higher speedups and additional memory savings. Moreover, the requirements on these two limits differ between the forward and backward passes due to the sparse-matrix transposition used in backpropagation. We note that relevant future extensions could characterize these requirements and develop online tuning of these hyperparameters to improve performance and memory savings further. Finally, when the number of elements exceeds the capacity of our data structures, we currently discard the excess values to avoid a hard failure and set a flag that is reported to the CPU at the next GPU synchronization point. This allows the training system to adaptively increase the structure sizes and repeat the latest training optimization step to avoid any deterioration in the learning dynamics.

*Table 4.* Comparison of performance and efficiency statistics of sparse LLMs leveraging our kernels with traditional baselines, considering both gated models (Shazeer, 2020), and their original non-gated counterparts used in the original transformer (Vaswani et al., 2017).

| Model scale | Gated | Sparse | L1 coefficient | Mean task accuracy | Forward execution (input tokens/ms) | | Energy per token (mJ) | |
|---|---|---|---|---|---|---|---|---|
| 1.5B params 30B tokens | ✓ | ✗ | 0 | 46.36% | 117.1 | (0.0%) | 5.79 | (0.0%) |
| | | ✓ | $2 \times 10^{-5}$ | 46.20% | 138.0 | (+17.9%) | 5.07 | (-12.5%) |
| | | ✓ | $3 \times 10^{-5}$ | 44.83% | 147.0 | (+25.5%) | 4.75 | (-18.0%) |
| 1.5B params 30B tokens | ✗ | ✗ | 0 | 46.57% | 125.8 | (0.0%) | 5.52 | (0.0%) |
| | | ✓ | $2 \times 10^{-5}$ | 46.46% | 139.9 | (+11.2%) | 5.03 | (-8.8%) |
| | | ✓ | $3 \times 10^{-5}$ | 44.71% | 142.3 | (+13.1%) | 4.86 | (-12.0%) |

# C. Parameter Studies and Ablations

## C.1. Performance and Efficiency Across Activation Functions

As noted in Section 2, many recent LLM architectures have deviated from using ReLUs in favor of smoother activation functions such as SiLU (Hendrycks, 2016; Ramachandran et al., 2017). To provide a direct comparison between the two activations, we collect additional training runs on 30B tokens with our 1.5B model and collect efficiency and performance results. In Table 3, we find that, while final cross-entropy appears equivalent, SiLU activations indeed yield slightly higher task accuracy in our evaluation set. However, we note that SiLU LLMs are already marginally slower than non-sparse ReLU LLMs by 0.5%, and due to their inherent non-sparse nature, they cannot support integration with sparsity and, therefore, the benefits of our kernels. Overall, we find these results are consistent with the ones from (Mirzadeh et al., 2023) using larger OPT models (Zhang et al., 2022a) – appearing to indicate that the advantages of smooth activation functions are minor and could potentially be offset by efficiency considerations.

## C.2. Non-gated Sparse LLMs

As explained in Section 2, from the simple 2-layer feed-forward block used in the original transformer, there has been a notable shift, with modern models adopting a gated variant due to small but consistent superior empirical results. Nonetheless, in our work, we introduce training and inference kernels for both variants. In contrast to the gated variant, computing the output activations following 1, for the non-gated variant, the computation simplifies to:

$$h = \phi(xW_u), y = hW_d, \tag{5}$$

where $\phi$ is, once again, the non-linear activation function. Thus, when $\phi$ is a ReLU activation, the sparsity pattern is determined by the up-projection rather than the gate projection. For inference kernels, we note this implies that a difference between the two variants is that the non-gated version performs the up projection rather than the gate projection with our matrix multiplication kernel with TwELL storage introduced in Section 3. Moreover, as detailed in Appendix A, we designed an additional kernel optimized to perform the down projection alone starting from the TwELL format.

Thus, to provide a direct comparison between the two variants and the relative effects of sparsity and our custom kernels, we collect additional training runs on 30B tokens with our 1.5B model implementing the non-gated parameterization. In particular, we consider two sparsity levels in addition to a non-sparse baseline ($L_1 = 0$): our recommended conservative regularization of $L_1 = 2 \times 10^{-5}$ and a more aggressive regularization of $L_1 = 3 \times 10^{-5}$. In Table 4, we report the collected relative performance and efficiency results for both variants and all three sparsity levels. As shown, we find only minor performance differences between the two variants, which are likely not significant and attributable to random variations. However, we do note that such differences might become visible only when training with token budgets beyond chinchilla optimality (Hoffmann et al., 2022). Efficiency-wise, while both our variants benefit significantly from our sparse kernels, we find such benefits to be larger for the gated variant. The inference speedup of the non-gated variant is 11.2% at $L_1 = 2 \times 10^{-5}$ compared to 17.9% for the gated variant at the same sparsity level. At larger sparsity levels, this divide is more pronounced, with the gated variant achieving a 25.5% speedup at $L_1 = 3 \times 10^{-5}$ compared to only 13.1% for the non-gated variant. These results are intuitively based on the nature of both models, as the gated variant allows our new inference kernels to leverage the opportunity of efficient fast fusion of both up and down projections. Nonetheless, they also demonstrate that the benefits of our kernels extend beyond a single architectural choice.

*Table 6.* Comparison of performance and efficiency statistics of sparse LLMs leveraging our kernels with traditional baselines trained using our standard recipe, or with dead neuron mitigation strategies such as warming up the coefficient of the L1 loss and applying targeted reinitialization to the gate projection's weights.

| Model scale | Training modification | | Sparse L1 coefficient | Mean task accuracy | Cross-entropy | # non-zeros | Forward execution (input tokens/ms) | | Energy per token (mJ) | |
|---|---|---|---|---|---|---|---|---|---|---|
| **1.5B params** 30B tokens | – | ✗ | 0 | 46.4% | 2.255 | 911 | 117.1 | (0.0%) | 5.77 | (0.0%) |
| | – | ✓ | $2 \times 10^{-5}$ | 46.2% | 2.297 | 29 | 138.0 | (+17.9%) | 5.07 | (-12.1%) |
| | Dead neuron reinit. | ✓ | $2 \times 10^{-5}$ | 46.6% | 2.298 | 29 | 139.4 | (+19.1%) | 4.96 | (-14.0%) |
| | Sparsity warmup | ✓ | $3 \times 10^{-4}$ | 45.9% | 2.293 | 108 | 119.3 | (+1.9%) | 5.76 | (-0.1%) |

## C.3. Combining Structured and Unstructured Sparsity

We also collected preliminary experiments confirming the complementarity between structured and unstructured sparsity, validating that our sparse kernels transfer to Mixture-of-Experts models (MoE) (Shazeer et al., 2017; Lepikhin et al., 2020; Fedus et al., 2022). Here, we simply apply L1 regularization together with our kernels independently for each individual expert. In Table 5, we report our results for MoE models at 1.5B and 2.5B total parameters. Interestingly, we found that these models require a much smaller regularization strength with $L_1 = 3 \times 10^{-6}$

*Table 5.* Downstream performance and efficiency results after extending sparse training to Mixture-of-Experts models and applying our inference kernels with TwELL storage in every expert.

| Model scale | L1 coeff. | Accuracy sparse / baseline | Speedup | Energy savings |
|---|---|---|---|---|
| **MoE-1.5B params** 1B active / 30B tokens | $3 \times 10^{-6}$ | 42.5% / 42.5% | 12.7% | 11.0% |
| **MoE-2.5B params** 1.5B active / 30B tokens | $3 \times 10^{-6}$ | 45.3% / 45.1% | 14.4% | 15.6% |

being enough to reach comparable levels of sparsity of our original $L_1 = 2 \times 10^{-5}$ for standard LLMs. Given the smaller scale of these preliminary experiments, compared to large MoEs actually trained in practice, we purposefully kept a high number of active parameters to avoid underutilizing our GPUs and still achieving significant performance. Analogously with our other results, we expectedly found that both MoE models preserved downstream accuracy relative to their non-sparse baselines while obtaining visible speedups and energy savings with our inference kernels. Moreover, the scaling model size again seems beneficial, with the 2.5B model obtaining 1.7% higher speedups and 4.6% more energy savings compared to its smaller counterpart, something of particular relevance given the scale of MoE models used in modern production environments.

## C.4. Alternative Activation Functions

We also run preliminary experiments training new non-gated LLMs with 1.5B parameters, and ReLU squared non-linearities (So et al., 2021). To run these experiments, we adapted our inference kernels, leveraging the TwELL data format to support ReLU squared non-linearities in the feed-forward blocks. After training, we collected results comparing downstream accuracy, throughput speedups, and energy savings against our non-gated LLM baselines at two different sparsity levels. In line with the results from prior work (Zhang et al., 2024), we found that ReLU squared LLMs are slightly more performant than their regular non-gated counterparts, increasing downstream accuracy from 46.5% to 47% at our conservative $L_1 = 2 \times 10^5$. Interestingly, we also found that ReLU squared models are slightly less sparse on average ( 10% more non-zeros), but also that such sparsity is more uniformly distributed across different LLM layers. As shown in Figure 10, thanks to this latter property, we found that our fused kernels provide marginally higher throughput gains than in regular non-gated LLMs. We believe that these additional results provide more insights into the role of activation functions and highlight future directions for more efficient LLM sparsity.

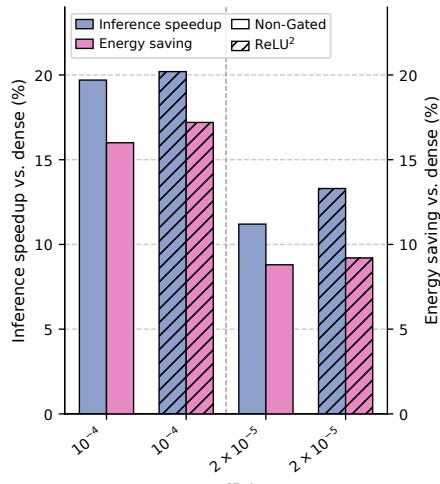

*Figure 10.* Forward execution speedups and energy savings for non-gated ReLU and ReLU squared LLMs across L1 regularization levels.

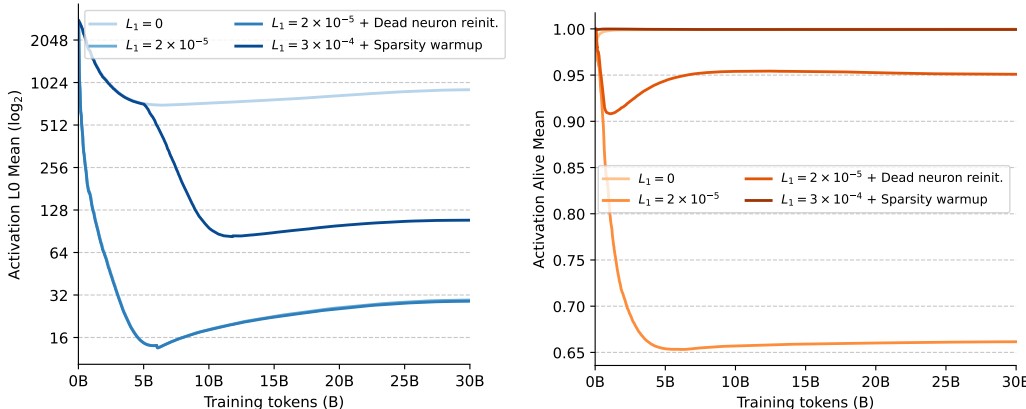

*Figure 11.* Number of non-zeros and fraction of dead neurons of LLMs with different strategies for dead neuron mitigation throughout training.

### C.5. Strategies for Dead Neuron Mitigation

While we find that using an L1 coefficient of $L_1 = 2 \times 10^{-5}$ provides a relevant boost in efficiency without any noticeable downstream performance degradation, we explore preliminary directions to mitigate the potential downsides of sparse training. In particular, as detailed in Appendix D, when examining the number of active neurons throughout training, we see that over 30% of the neurons become permanently inactive on average across layers when using our recommended L1 coefficient, with this metric considerably rising for higher regularizations. While for our recommended coefficient, this symptom does not seem to evidently reflect on downstream performance, reducing such an effect could potentially allow supporting even higher sparsity before incurring performance degradation.

Based on these considerations, we explore two preliminary extensions to our simple L1-regularized training recipe explained in Section 2. First, we consider simply scheduling the L1 regularization, motivated by our findings that dead neurons appear to arise very early during training. Concretely, we first train our models for 5000 steps without any L1 regularization, followed by a further 5000 steps of linear increase of the L1 coefficient. We make the training setting artificially similar to prior work that focuses on finetuning and continued-pretraining (Song et al., 2025; Wang et al., 2024). Second, we consider implementing a target reinitialization strategy to lower the magnitude and reinject random noise only in the columns of the gate projection that lead to always negative outputs (which then, after ReLU, lead to dead neurons). Given the model's initialization standard deviation $\sigma = 0.02$, we noised and rescaled to regress the weights to their initial state, essentially interpolating with a coefficient $\lambda$:

$$W_g[:, j] \leftarrow (1 - \lambda)W_g[:, j] + \lambda \mathcal{N}(0, \sigma^2), \tag{6}$$

We apply this targeted reinitialization after every training step, which we find does not significantly affect training time. In preliminary experiments, we found $\lambda = 0.1$ to be a good choice that avoids affecting training dynamics while injecting sufficient noise to revive dead neurons. We note this strategy is similar to older techniques for reinjecting plasticity into architectures in continual learning and other non-stationary settings (Ash & Adams, 2020).

In Table 6, we report the performance and efficiency results of our two strategies compared to our standard recipe and the non-sparse baseline, while in Figure 11 we analyze the number of non-zero activations and dead neurons throughout training. When looking at the dead neuron statistics, we find that both strategies almost entirely mitigate the emergence of dead neurons. However, we immediately see a concerning pattern with the sparsity-warmup strategy, as the number of non-zeros considerably increases throughout training. In particular, the considered coefficient of $L_1 = 3 \times 10^{-4}$, which is ten times larger than our recommended value, leads to over 100 non-zeros on average across layers at the end of training, compared to only 29 non-zeros when using our standard recipe with $L_1 = 2 \times 10^{-5}$. We note that, in early experiments, we found that increasing the L1 coefficient further led to training instabilities and loss spikes. In contrast, using the targeted dead neuron reinitialization, we find similar non-zero statistics to our standard recipe while still effectively mitigating dead neurons. Furthermore, as reported in Table 6, we find that this latter strategy provides a small boost in both downstream performance and efficiency, processing tokens 19.1% faster than the non-sparse baseline with our default L1 coefficient of $L_1 = 2 \times 10^{-5}$. We believe these preliminary results suggest that further research in examining optimal sparse training would potentially further increase the relevance and efficiency upsides of sparse LLMs.

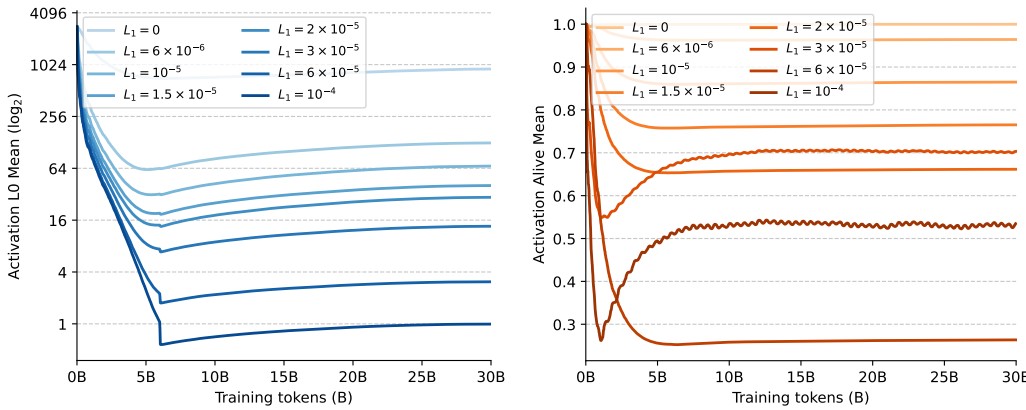

*Figure 13.* Number of non-zeros and fraction of dead neurons of LLMs across L1 regularization levels throughout training.

## C.6. Sparsity and Continual Pretraining

We also run preliminary experiments to reproduce the results from prior work (Mirzadeh et al., 2023), to validate that modern LLMs can be sparsified without significant losses in downstream accuracy. In particular, our experiments focus on the Qwen-2.5-7B base model (Yang et al., 2024), which is significantly larger than all our other models, used SiLU activations (Hendrycks, 2016; Ramachandran et al., 2017), and was pretrained with a proprietary closed-source dataset. To sparsify this model, we swapped SiLU activations with ReLU in Qwen-2.5-7B and finetuned the model with mild L1 loss of $6 \times 10^5$ for 20B tokens using data from Nemotron pretraining dataset (Basant et al., 2025). To control costs, we opted to lower the maximum context size to 4096. As summarized in Figure 12, and in line with the results in (Mirzadeh et al., 2023), our results indicate that LLMs can recover over 95% of their downstream performance after only 20B tokens. Moreover, after training, we adapted and applied our inference kernels to the Qwen model, resulting in non-trivial throughput speedups, up to 9.6%, and energy savings, up to 8.9%. We hope that future work will expand these

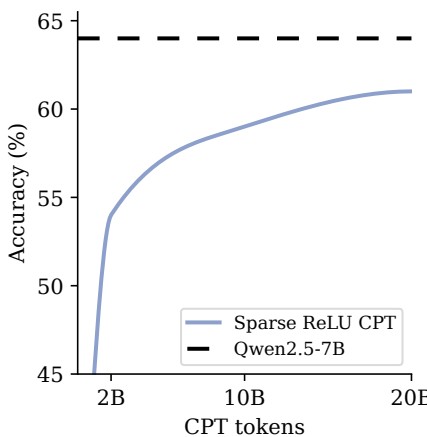

*Figure 12.* Downstream accuracy recovery after swapping SiLU activations with ReLU in Qwen2.5-7B and applying sparse continual pretraining.

preliminary results with more extended continued pre-training to further lower performance gaps, and also investigate sparsification strategies during and after post-training, which is becoming increasingly expensive with modern LLMs.

## D. Extended Results

### D.1. Sparsity and Dead Neurons During Training

In Figure 13, we provide detailed results about how activation sparsity and dead neuron occurrence evolve during training for all our different L1 regularization levels. In particular, we record dead neurons at the end of each training step by keeping track, for each hidden feedforward activation of each layer, the last time it was non-zero. If a neuron was never active for a whole training step (just above 1M tokens), we consider it dead for that step.

We make two immediate observations from these results. First, we find that the sparsity levels settle early on to low values after only around 1000 training steps (around 1B tokens). Due to this property, we note that the throughput and memory advantages of our training kernels become relevant almost at the inception of our training runs. Second, we observe that the same trend applies to the number of dead neurons: our recommended $L_1 = 2 \times 10^{-5}$ already exceeds 30% inactivity, which further monotonically increases with higher regularization levels. While for our recommended coefficient, this symptom does not seem to evidently reflect on downstream performance, reducing such an effect could potentially allow supporting even higher sparsity before incurring performance degradation. To this end, we note that in Appendix C we provide preliminary results indicating that the performance of sparse LLMs could be further improved with strategies targeted at dead-neuron mitigation.

*Table 7.* Granular comparison of per-task downstream performance across model scales to complement Table 1.

| Model scale | Sparse | Mean Accuracy | HellaSwag | CQA | PIQA | Winogrande | ARC-easy | ARC-challenge | OpenBookQA |
|---|---|---|---|---|---|---|---|---|---|
| 0.5B params
10B tokens | ✗ | 40.4% | 33.7% | 20.9% | 64.5% | 50.9% | 64.1% | 28.1% | 20.8% |
|  | ✓ | 40.4% | 34.0% | 22.3% | 66.4% | 53.8% | 60.5% | 27.5% | 18.0% |
| 1B params
20B tokens | ✗ | 44.6% | 39.2% | 20.0% | 68.7% | 54.4% | 72.6% | 34.0% | 23.6% |
|  | ✓ | 44.7% | 39.8% | 18.6% | 68.1% | 54.8% | 71.6% | 35.3% | 24.4% |
| 1.5B params
30B tokens | ✗ | 46.4% | 41.0% | 20.8% | 70.2% | 55.9% | 72.5% | 36.7% | 27.4% |
|  | ✓ | 46.2% | 41.1% | 21.0% | 69.1% | 54.4% | 74.3% | 37.5% | 26.0% |
| 2B params
40B tokens | ✗ | 49.1% | 45.7% | 21.0% | 72.0% | 57.8% | 77.2% | 41.7% | 28.6% |
|  | ✓ | 48.8% | 45.0% | 21.3% | 70.9% | 57.5% | 75.6% | 42.2% | 28.8% |

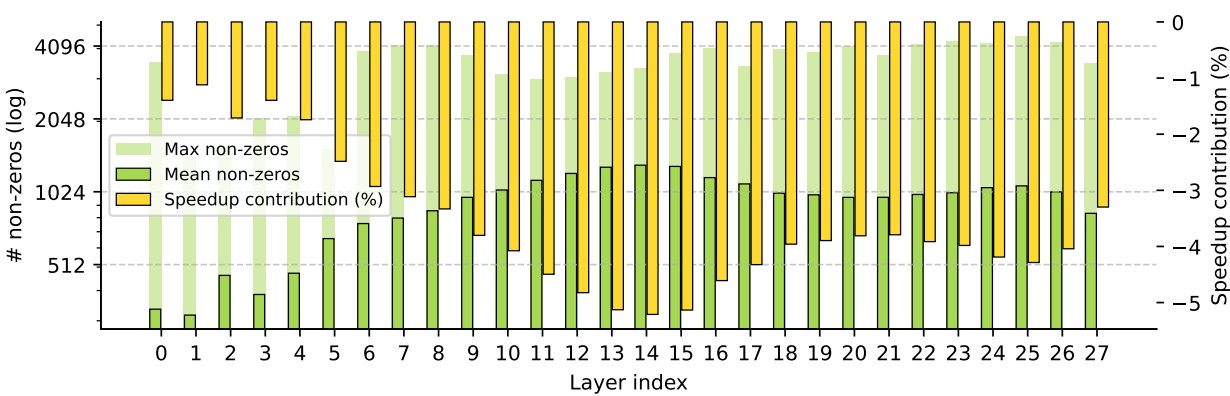

*Figure 14.* Sparsity statistics and speedup contributions across different layers of non-sparse LLMs.

## D.2. Task Performance Details

To complement the results in Section 4 in the main text, we provide the detailed granular results of downstream task performance across the seven downstream tasks considered, targeting logic and reasoning capabilities after pretraining (Clark et al., 2018; Zellers et al., 2019; Mihaylov et al., 2018; Bisk et al., 2020; Sakaguchi et al., 2021; Talmor et al., 2019). In particular, we report the per-task accuracies for both sparse models, using our recommended conservative L1 regularization of $2 \times 10^{-5}$, and their non-sparse counterparts across all the examined model scales. As shown in Table 7 and consistently with our main text analysis, we do not find significant performance differences between sparse and non-sparse models for our regularization level and all considered tasks. We do, indeed, observe an expected performance rise with larger models across the great majority of tasks.

## D.3. Activation Sparsity at High and Low Levels

To complement the analysis results provided in Section 4 of the main text, we examine how sparsity regularization affects the distribution of non-zero activations across model depth and relate these metrics to the corresponding speed-up contributions from our kernels during inference. While in our main analysis we reported and analyzed the LLM trained with our recommended conservative L1 regularization of $2 \times 10^{-5}$, in Figures 14 we provide analogous results for a non-sparse LLM while in Figure 15 we analyze an LLM trained with the highest regularization regularization level considered ($1 \times 10^{-4}$). We note that for non-sparse models, due to the high number of non-zeros, the contributions of applying our kernel are actually detrimental – and as such, we report the speed-up contributions as negative percentages. A first observation from the sparsity statistics is that the average number of non-zeros

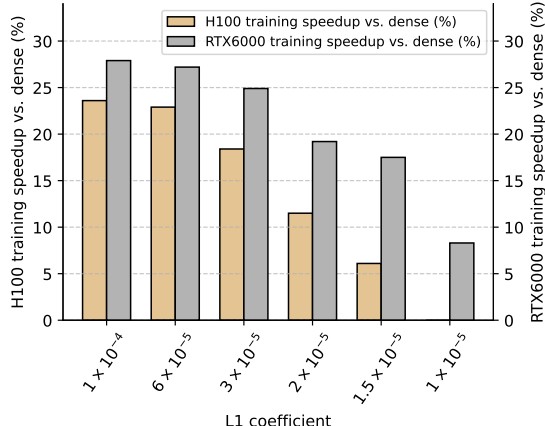

*Figure 16.* Training speedups from our sparse LLM training kernels across L1 regularization levels for both H100 and RTX6000 devices.

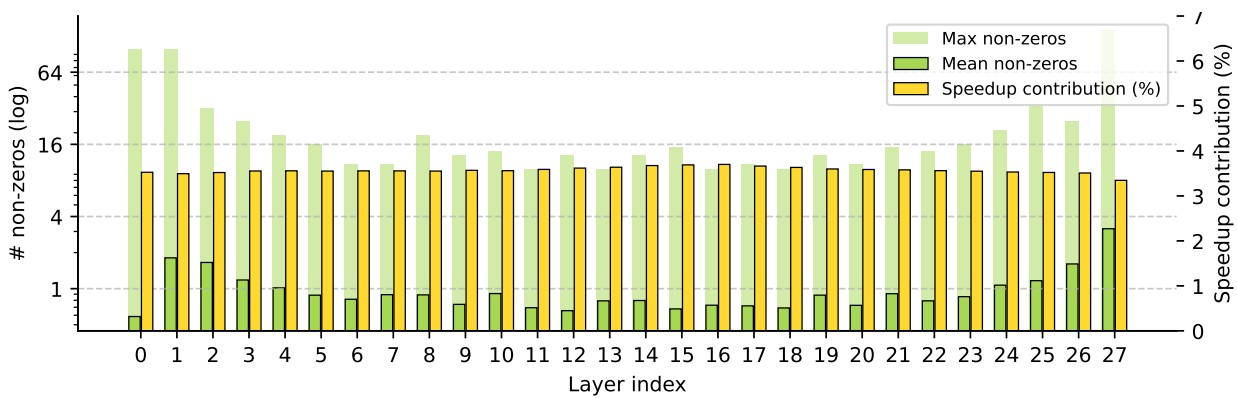

*Figure 15.* Sparsity statistics and speedup contributions across different layers of an LLM with high regularization $L_1 = 10^4$.

*Table 8.* Head-to-head comparison between our kernels and sparse inference baselines in batched GEMM regimes. Higher throughput is better.

| L1 regularization | Inference throughput (tokens/ms) | | | | | |
|---|---|---|---|---|---|---|
| | Torch | ProSparse | TEAL | TEAL++ | ELL-R++ | TwELL |
| $2 \times 10^{-5}$ | 114.7 (0.0%) | 5.3 (-95.4%) | 11.6 (-89.9%) | 58.8 (-48.7%) | 100.0 (-12.8%) | **141.1** (+23.0%) |
| $3 \times 10^{-5}$ | 114.5 (0.0%) | 5.4 (-95.3%) | 11.6 (-89.9%) | 60.0 (-47.6%) | 108.0 (-5.7%) | **146.2** (+27.7%) |
| $1 \times 10^{-4}$ | 115.0 (0.0%) | 5.4 (-95.3%) | 11.7 (-89.8%) | 61.0 (-47.0%) | 120.3 (+4.6%) | **152.2** (+32.3%) |

also follows a noticeable trend in the non-sparse model, with the first few layers being the least active, followed by a hump with a peak in activations. However, a key difference comes with the location of the hump: while in our recommended sparse model the peak occurs around layer 6, in the non-sparse LLM the peak still occurs within the first half of the network but is shifted visibly deeper into the architecture around layer 13. Interestingly, in the high-regularization LLM, we actually observe that while the very first layer is again the least active, there are two different peaks – one very early around the second layer and another one in the last layer of the model. Once again, we find that maximum activation counts can easily be well over an order of magnitude higher than the average, with no clear pattern across layers. For the non-sparse model, we again observe a strong inverse correlation between each layer's average non-zeros and its relative speed-up. In contrast to the high-regularization LLM, this correlation is much less visible, as given the high sparsity encountered, the speedups of our kernels are already at their achievable maximum for almost all layers, essentially making executing the up and down projection negligible in the overall computation time.

### D.4. Adapting and Comparing Prior GEMV Kernels

As detailed in Section 5 and Appendix E, prior works focusing on sparsity, such as DejaVu (Liu et al., 2023), TEAL (Liu et al., 2024), CATS (Lee et al., 2024), and ProSparse (Song et al., 2025), mainly target GEMV operations for non-batched inference settings. In these cases, the dense baseline kernels tend to already underutilize compute on the GPU, without leveraging Tensor Cores. In these regimes, the execution of the feedforward modules is entirely memory-bound. This is in contrast to our compute-bound batched setting, where the dense baselines fully utilize Tensor Cores, and relying on kernel fusion and higher sparsity levels is crucial for obtaining speedups. To validate these inherent differences, we port several kernels from prior work to our batched setting, and provide preliminary results in Table 8. In particular, we integrate the kernels from TEAL and ProSparse into our batched LLM inference benchmark, removing batch size restrictions, and use them in the down projections of our pretrained models. Moreover, we also evaluate more GEMM-specific variants by replacing only the core logic of our optimized TwELL kernels with prior techniques, resulting in two additional baselines: ELL-R++, a baseline which uses separate data conversion followed by a sparse matmul with the ELLPACK-R format (Vazquez et al., 2010), and TEAL++, which uses the dynamic activation thresholding present from the TEAL kernels (Liu et al., 2024) rather than sparse packing. As these prior methods were not developed for fully batched inference, our results intuitively validate that, without explicit sparse packing, kernels optimized for GEMV are not competitive in fully batched regimes. Moreover, by reducing materialization costs, our TwELL kernels yield much higher throughput compared to prior ELL-based formats.

*Table 9.* Performance and sparsity statistics after applying GPTQ weight quantization to sparse and non-sparse models.

| L1 regularization | Mean task accuracy | | | Final cross-entropy | | | Number of non-zeros | | |
|---|---|---|---|---|---|---|---|---|---|
| | 16-bit | 8-bit | 4-bit | 16-bit | 8-bit | 4-bit | 16-bit | 8-bit | 4-bit |
| 0 | 46.4% | 46.5% | 46.5% | 2.236 | 2.236 | 2.251 | 911 | 911 | 910 |
| $3 \times 10^{-6}$ | 47.0% | 47.1% | 47.0% | 2.237 | 2.237 | 2.251 | 241 | 241 | 241 |
| $2 \times 10^{-5}$ | 46.1% | 46.1% | 45.2% | 2.295 | 2.295 | 2.322 | 29.4 | 29.4 | 29.3 |

### D.5. Combining Sparsity and Weight Quantization

We add preliminary results to validate that activation sparsity and weight quantization do not have any adversarial interactions with each other and can act as complementary techniques to increase efficiency. To test this, we apply GPTQ weight quantization (Frantar et al., 2022) to sparse and non-sparse models, compressing the weights of our 1.5B model for both 8-bit and 4-bit precision. In Table 9, we then compare downstream task accuracy, cross-entropy, and the number of non-zero activations at the end of training. Our result confirms that quantized sparse and non-sparse models appear to behave analogously, with 8-bit quantization preserving all reported metrics, while 4-bit quantization produces only extremely minor deviations in final cross-entropy and downstream accuracy, irrespective of the regularization level.

### D.6. Improving Efficiency of other Devices

As mentioned in Section 4, given that our kernels consistently reduce memory requirements during training, and as a side benefit, reduce reliance on newer tensor core units, they immediately have higher potential relevance for less capable hardware. Thus, to empirically validate these considerations, we provide additional results comparing the performance speedups of our kernels during training on NVIDIA's RTX PRO 6000 GPUs against the H100 PCIe GPUs used throughout our main paper and other experiments. Some of the other crucial differences of this GPU come from the memory side, with a considerably reduced memory bandwidth (1.59 TB/s vs. 2.0 TB/s). In contrast, the RTX PRO 6000 can benefit from a larger number of Streaming Multiprocessors than the H100 (188 vs. 114), potentially allowing for greater occupancy for sparse workloads.

As shown in Figure 16, and in line with our considerations, we find significantly higher speedups on the RTX 6000 GPU across all L1 regularization levels considered. These speedup differences are even more pronounced at higher regularization levels, extending the practical range of L1 coefficients, making sparsity provide meaningful efficiency improvements. When dissecting what causes these greater speedups, we first find that thanks to the specific H100 features, such as the higher tensor cores throughput, the runtime of the dense GEMM operations increases from around 400 to 800 microseconds on the RTX 6000. Similarly, kernels that are memory bandwidth bound, including the dense to hybrid matrix multiplication, are also slightly slower by 19% on the RTX 6000 than on the H100. However, once in our hybrid sparse format, due to the larger Streaming Multiprocessors count of the RTX 6000 GPU, the sparse operations run faster than on the H100, with speedups of $1.34\times$ and $2.1\times$ for sparse-to-dense and transposition operations, respectively. We find these results indicate that leveraging sparsity with targeted kernels could significantly improve the performance of cheaper devices, which do not implement the latest hardware innovations of higher-end units such as the H100, lowering the field's canonical hardware barriers.

## E. Further Related Work

### E.1. Activation Sparsity in Transformers

Expanding on the findings of Zhang et al. (2022b), Li et al. (2023) documents that Transformer MLP layers with ReLU activations exhibit inherent activation sparsity across architectures, depths, and data distributions. Building on this observation, Mirzadeh et al. (2023) shows that replacing GELU with ReLU in non-gated feed-forward layers yields negligible performance degradation while enabling up to three times theoretical inference speedup with less computation. However, they focus on older architectures (OPT models) with non-gated feed-forward blocks and leave efficient kernel implementation to future work.

More recent methods have also been proposed to enhance sparsity after altering modern gated architectures and have claimed speedups when running sparse feedforward layers in isolation on older generations of devices. TurboSparse (Song et al., 2024) proposes a modification to the feed-forward block itself, introducing dReLU, which applies ReLU to *both*

gate and up projections: $h = \text{ReLU}(xW_g) \odot \text{ReLU}(xW_u)$. ProSparse (Song et al., 2025) proposes finetuning pretrained models and artificial thresholding of the activations to increase sparsity. Q-Sparse (Wang et al., 2024) further deviates from standard architectures via maintaining only the top-K activations and applying a straight-through estimator. We also note that additional works proposed introducing sparsity post-training, such as by predicting (Liu et al., 2023) and pruning activation to set sparsity levels (Lee et al., 2024; Liu et al., 2024). However, together with other related efforts (Macko & Boža, 2025), the kernels employed to leverage unstructured sparsity from prior research mainly target memory-bound GEMV operations for the single/few-token regime. Unlike these prior works, our paper introduces general-purpose kernels for compute-bound GEMM operations in batched settings with thousands of input tokens, where dense baselines on modern devices can execute up to orders-of-magnitude higher FLOP/s with large tiles and Tensor Cores. Moreover, focusing on the batched setting allows our work to demonstrate that leveraging unstructured sparsity can provide empirical efficiency benefits during both LLM inference and training.

### E.2. Architectural Approaches to Sparsity

Mixture-of-Experts (MoE) architectures (Shazeer et al., 2017; Lepikhin et al., 2020; Fedus et al., 2022) partition feed-forward layers into separately routed experts, decoupling model capacity from per-token computation. However, MoE requires predetermining the number of experts and sparsity level before training, limiting adaptability to input complexity.

Product key memory (Lample et al., 2019) maintains fixed sparsity patterns through $O(\log n)$ key retrieval. PEER (He, 2024) extends this approach to over one million single-neuron experts with 99.99% architectural sparsity. UltraMem (Huang et al., 2025) improves PKM and scales to 20 million memory slots, showing that it can outperform MoE with the same parameter and computation budgets. Fast Feedforward Networks (Belcak & Wattenhofer, 2023) use differentiable binary trees to achieve 99% sparsity.

While these architectural approaches achieve extreme sparsity, they require substantial modifications to standard Transformer training pipelines. Our approach instead works with conventional architectures, requiring only a change of activation function and optional regularization, making it readily applicable to existing models and training infrastructure.

### E.3. Seminal and Theoretical Research

Prior work focusing on convolutional neural networks for computer vision has successfully applied L1 regularization to "network compression" to achieve model size reduction. For instance, Filters'Importance (2016) proposed pruning convolutional filters ranked by their L1-norm, with Liu et al. (2017) adding an L1 penalty to identify and remove redundant channels. On the theoretical side, the use of L1 regularization to induce sparsity is well-grounded in classical statistical learning theory. The geometry of the L1 constraint (a polytope whose vertices lie on coordinate axes) causes solutions to concentrate at corners where coefficients are exactly zero (Ruppert, 2004). More recently, Bach (2017) showed that an L1-type penalty on single-hidden-layer ReLU networks yields generalization bounds that can be independent of input dimensionality, and Trauger & Tewari (2024) established generalization bounds for full transformer architectures in which $\ell_{1,\infty}$ norms on weight matrices appear as the controlling complexity measure.

