# OpenReview forum: "Sparser, Faster, Lighter Transformer Language Models"
_ICML.cc/2026/Conference — ICML 2026 regular_

### Official Review · Reviewer_uQNv · 2026-02-17

**Soundness:** 2
**Presentation:** 1
**Significance:** 3
**Originality:** 3
**Overall Recommendation:** 4
**Confidence:** 2

**Summary:**

This paper introduces a novel method on efficiently using the unstructured sparsity in FFNs during training and inference, to achieve a faster inference speed without significantly harming the performance of LLM. Such method adopts a new sparse matrix representation, TwELL, that stores the non-zero elements in values, offsets, and sizes with matrix tiling. To achieve a higher sparsity, this paper introduces an L1-norm loss on the activations of the FFN. TwELL is compatible in both training and inference that minimizes the overhead of matrix multiplication and backwarding. Empirical results are reported to show the performance-efficiency tradeoff of using TwELL with additional L1 loss, where ~1.2-1.3x speedups are achieved without notable performance loss. The authors also provide Triton-implemented kernels of FFN using TwELL.

**Compliance With Llm Reviewing Policy:**

Affirmed.

**Final Justification:**

All questions resolved.

**Key Questions For Authors:**

1. Could you explain the reason why there is an increase of the peak memory in Table 1's 2B setting?

2. How efficient are the kernels? If possible, measuring and reporting the largest achievable floating point operations per second (FLOPs) under various sparsity would an appretiated response.

**Limitations:**

yes

**Strengths And Weaknesses:**

## **Strengths**
---

1. The TwELL sparse matrix representation is novel,  fundamental, and effective. Storing the non-zero elements with matrix tiling provides a good layout for acceleration in matrix multiplication, and sparse training with such format is fully differentiable. Such contribution is not only useful for future works that explores the unstructured sparsity in FFNs, but also provides a novel method on simply speeding up matrix multiplication.

2. Experiments on multi-scale models shows the scalability of the introduced method. Table 1 shows that the efficiency gains grow when scaling the model size, while the performance losses remains low and acceptable.

3. This paper also explores energy saving using the TwELL representation, and Figure 4, 5 show positive results. Such improvement of energy consuption unlocks more scenarios of using TwELL that requires low energy cost, e.g., edge side LLM deployment where the power supplement is not sufficient or stable.

## **Weaknesses**
---
1. Although this paper includes a large number of related works in Section 5, non of them appears in the experiment section as a baseline. This paper positions itself as an "unstructured sparsity" method and may not be similar with structured sparsity methods and MoE methods, but neglecting these methods as experimental baselines hinders readers from acknowledging the real potential of TwELL in performance-efficiency tradeoffs. My suggestion is to do at lease one of the following choices: a) pick one or two methods, such as ProSparse [1] and Q-Sparse [2], and compare the performance under same-level sparsity; b) integrate TwELL into some structured sparsity / MoE methods to show the orthogonality of TwELL and existing methods. These extra results can make the **soundness** of this paper better.

2. The **presentation** of this paper can be improved. Instead of listing the Algorithm 1 and 2 (as well as the Triton codes in the appendix), the algorithms should be presented in a clearer way, e.g., using diagrams. Also, please use standard math notations and avoid list-like references (e.g.  `h[m, n]`) in formal equations.

3. The title of this paper seems to be too broad, as readers cannot grasp the main contributions of this paper easily. Since the sparsity is only explored in FFNs, another large component, Attention, remains untouched in the transformer-based LLMs. It would be better to use a more specific title.

---

[1] Prosparse: Introducing and enhancing intrinsic activation sparsity within large language models

[2]  Q-sparse: All large language models can be fully sparsely-activated.

---

> ### Author Rebuttal · Authors · 2026-03-30
>
> We thank the reviewer for their time and constructive feedback.
>
> **Baselines**
>
> Based on the reviewer’s feedback, **we added comparisons under same-level sparsity** for sparse matmul kernels and **extended Related Works** to more clearly convey differences with prior work:
>
> Our work introduced new sparse formats and GEMM kernels for modern GPUs that outperform dense kernels in training and batched inference settings (where we select the batch size to fill our GPU’s HBM). In contrast, the kernels in ProSparse and other related work, such as DejaVu [1] and TEAL [2], only target GEMV operations for non-batched inference, where the dense torch baseline already underutilizes compute without using TensorCores (i.e., [where few/a single token is processed at a time](https://github.com/FasterDecoding/TEAL/issues/12)).
>
> Thus, as recommended by the reviewer, **we added two kinds of head-to-head comparisons** to validate these differences:
>
> First, we integrated TEAL and ProSparse into our batched LLM inference to compare throughput. We also evaluated more GEMM-specific methods by replacing only the core logic of our optimized TwELL kernels with prior techniques, resulting in two more baselines:
> - ELL++: A baseline using separate data conversion and sparse matmul with the ELLPACK-R format [3] introduced in Sec 3
> - TEAL++: A baseline using dynamic activation thresholding (from the TEAL kernels) rather than sparse packing
>
> |GEMM(tok/ms)|2e-5|3e-5|1e-4|
> |-|-:|-:|-:|
> |Torch|114.7|114.5|115.0|
> |ProSparse|5.3|5.4|5.4|
> |TEAL|11.6|11.6|11.7|
> |TEAL++|58.8|60.0|61|
> |ELL++|100|108|120.3|
> |TwELL|138.1|146.2|152.2|
>
> As summarized above, our results validate that kernels optimized for GEMV are not competitive for GEMM without sparse packing, and show the superiority of TwELL compared to past ELL formats.
>
> Second, since TwELL is not inherently limited to GEMM, we wrote a [sub-tiled extension of our ungated kernel for GEMV](https://anonymous.4open.science/r/sparse-anon-reb-5575/code_listings/matmul_t2d.cu). Thus, we reproduced the results in Fig 3 of the TEAL paper (comparing the DejaVu and TEAL kernels at different sparsity levels) and added TwELL results:
>
> |GEMV latency(µs)|50%|75%|90%|95%|99%|
> |:-|-:|-:|-:|-:|-:|
> |Theoretical|28.3|14.6|5.6|2.6|0.6|
> |Torch|55.7|55.8|55.6|55.8|55.7|
> |DejaVu|39.2|28.4|20.4|20.0|19.3|
> |TEAL|35.1|25.9|22.7|21.9|22.0|
> |TwELL|34.5|23.2|15.8|13.0|6.2|
>
> As summarized above, we find that TwELL can provide further GEMV speedups, especially visible with sparsity > 75%. We hope that these new comparisons will help contextualize the differences with prior work and our paper's contributions.
>
> **MoEs**
>
> Following the reviewer's suggestion, **we trained and evaluated sparse 1.5B and 2.5B MoEs**:
>
> |Model|L1|Acc(%, sparse/baseline)|Speedup(%)|Energy sav(%)|
> |:-|:-|-:|-:|-:|
> |MoE-2.5B|3e-6|45.3/45.1|14.4|15.6|
> |MoE-1.5B|3e-6|42.5/42.5|12.7|11.0|
>
> We refer to our second response to the first reviewer (HeU6) for a detailed analysis of these results, showing the orthogonal gains of structured and unstructured sparsity with TwELL.
>
> **Presentation**
>
> Based on the reviewer’s suggestions, **we added new diagrams to improve the clarity of Alg 1 and 2**, with a visual guide of costs by marking expensive DRAM ops:
>
> [The diagram for Alg 1 (see link)](https://anonymous.4open.science/r/sparse-anon-reb-5575/algorithm1_diagram.png) illustrates how the TwELL format is constructed from local packing of the activations’ tiles.
>
> [The diagram for Alg 2 (see link)](https://anonymous.4open.science/r/sparse-anon-reb-5575/algorithm2_diagram.png) provides a logical outline of kernel fusion between up and down-projections.
>
> **2B Model Peak Memory**
>
> Following the reviewer’s question, **we contextualized the 2B memory results more clearly by extending Sec 4.2 and the caption of Table 1**. We clarified that using our kernels to train all models greatly reduces memory requirements by over 30% for the 2B LLM. In fact, this reduction is what allows us to fit double the micro batch size (ln 374), which is precisely what causes the increase in peak memory, and results 21.9% faster training.
>
> **MLP FLOP/s**
>
> Following the reviewer’s second question, **we benchmarked the MLPs in our LLMs and compared FLOP/s of TwELL and dense Torch Kernels**. We report the mean and max-min range across layers to account for their differing sparsity statistics:
>
> |L1|Kernel|Mean(TFLOP/s)|Range(min–max)|
> |:-|:-|-:|:-|
> |Baseline|Torch|660|655-664|
> |2e-5|TwELL|1128|794-1546|
> |3e-5|TwELL|1448|1259-1691|
> |6e-5|TwELL|1745|1466-1788|
> |1e-4|TwELL|1819|1637-1842|
>
> As summarized above, our TwELL kernels notably increase FLOP/s with gains from 71% to 176% based on LLM sparsity. We hope these results will better highlight efficiency gains while ablating other confounding factors in LLM execution.
>
> [1] Liu et al Deja vu: Contextual sparsity for efficient llms
>
> [2] Liu et al Training-free activation sparsity
>
> [3] Vazquez et al Improving the performance of sparse matrix vector product

---

> > ### Author Rebuttal · Reviewer_uQNv · 2026-04-02
> >
> > Thanks for the timely review. I'm raising my overall to 4.

---

### Official Review · Reviewer_2i2g · 2026-02-24

**Soundness:** 3
**Presentation:** 2
**Significance:** 2
**Originality:** 3
**Overall Recommendation:** 4
**Confidence:** 4

**Summary:**

This paper proposes an LLM sparsification scheme based on ReLU and L1 regularization, which supports sparse LLM inference and training through the combination of TwELL and hybrid sparse formats. The authors conducted pre-training on the Transformer++ architecture with parameter scales from 0.5B to 2B, demonstrating their method can improve inference and training speed while reducing peak memory without compromising downstream task accuracy.

**Compliance With Llm Reviewing Policy:**

Affirmed.

**Final Justification:**

My initial score was 3 (Weak Reject). My primary concerns centered on the lack of head-to-head comparisons with related work and the method's effectiveness when appled to open sources pretrained LLMs.  The authors' rebuttal has satisfactorily addressed these issues, so I have updated my final score to 4 (Weak Accept).

**Key Questions For Authors:**

1. Could author provide head-to-head comparisons with existing methods?

2.  Would it be possible to further accelerate sparse inference by introducing TwELL on top of TEAL-like works? If not, what are the constraining factors?

3. Can the current method (ReLU + L1 regularization) maintain downstream task accuracy when modifying opensourced Qwen2.5-7B for CPT?

4. What impact would applying GPTQ/AWQ have on the models trained in this paper?

**Limitations:**

yes.

**Strengths And Weaknesses:**

Strengths:
1. Unstrutured sparsity is the most challenging area in current LLM acceleration. The authors provide a solution that combines unstructured sparsity with pre-training from a rigorous engineering perspective.

2. The authors provide complete code, which greatly enhanves the reproducibility of the proposed method.

Weaknesses:
1. While the authors have conducted solid work, the paper reads more like a technical blog than a conference submission. The Experimental Results section lacks head-to-head comparisons with existing methods, making it impossible to assess the effectiveness of the proposed sparsity acceleration approaches. The absence of comparatisons suggests that this paper **has no related work** in the unstructured sparsification research domain. I hope the authors can address this issue seriously.

2. The baselines used by the authors are somewhat outdated, which significantly undermines the contribution of this paper. I noticed in lines 137-145 that the authors attempt to cite some existing work to downplay the optimization benefits brought by SiLU activations, seemingly to avoid challenges to the necessity of ReLU in their proposed method. However, current mainstream LLMs and LRMs no longer contain ReLU modules, and algorithms such as TEAL, CATS, and Larosa have already demonstrated that ReLU modules are not essential for sparsification.

3. The authors need to demonstrate, either theoretically or experimentally, that their proposed inference and training algorithms are equally applicable to mainstream open-source models such as Qwen2.5-7B, Qwen3-8B, and Llama3-8B. Due to the enormous resource overhead of pre-training, the academic community primarily focuses on excellent open-source pre-trained models. However, since the authors' proposed training and inference pipelines both involve modifying model activations, this greatly limits the impact of this work.

4. The authors claim to propose an algorithm that combines sparsification with training to improve efficiency, which is a very good motivation. However, I need the authors to demonstrate the boundary conditions of this claim. For instance, when combined with DeepSpeed's stage zero 1, 2, 3, or with strategies such as PP/TP/CP in Megatron, to what extent would this algorithm be affected?

5. The authors do not demonstrate the orthogonality between their current method and PTQ schemes. What results would be obtained when applying weight-only or weight-activation PTQ algorithms to optimize the authors' sparse-trained models?

6. The paper lacks theoretical support regarding the effectiveness of L1-based sparsification. Based on my literature review, similar approaches were employed for model pruning in the CNN era -- 'Learning Efficient Convolutional Networks through Network Slimming'. It would be worthwhile to explore this line of research and include relevant citations.

---

> ### Author Rebuttal · Authors · 2026-03-30
>
> We thank the reviewer for their time and constructive feedback.
>
> **Baselines**
>
> Based on the reviewer’s feedback, **we added head-to-head comparisons** for sparse matmul kernels and **extended Related Works** to more clearly convey novelty and differences:
>
> Our work introduced new sparse formats and GEMM kernels for LLM training and inference on modern GPUs. Thus, our kernels outperform the throughput of dense compute-bound kernels in fully batched settings (where we select the batch size to fill our GPU’s HBM). In contrast, the kernels in prior work like DejaVu, TEAL, CATS, and Larosa mainly target GEMV operations for non-batched inference, where the dense torch baseline underutilizes compute without using TensorCores (i.e., [where few/a single token is processed at a time](https://github.com/FasterDecoding/TEAL/issues/12)).
>
> Thus, as recommended by the reviewer, **we added two kinds of head-to-head comparisons** to validate these differences:
>
> First, we integrated kernels in TEAL and Prosparse [1] into our batched LLM inference to compare throughput. We also evaluated more GEMM-specific methods by replacing only the core logic of our optimized TwELL kernels with prior techniques, resulting in two more baselines:
> - ELL++: A baseline using separate data conversion and sparse matmul with the ELLPACK-R format [2] introduced in Sec. 3
> - TEAL++: A baseline using dynamic activation thresholding (from the TEAL kernels) rather than sparse packing
>
> |GEMM(tok/ms)|2e-5|3e-5|1e-4|
> |-|-:|-:|-:|
> |Torch|114.7|114.5|115|
> |ProSparse|5.3|5.4|5.4|
> |TEAL|11.6|11.6|11.7|
> |TEAL++|58.8|60|61|
> |ELL++|100|108.0|120.3|
> |TwELL|138.1|146.2|152.2|
>
> As summarized above, our results validate that without explicit sparse packing, kernels optimized for GEMV are not competitive in fully batched regimes. Moreover, by trivializing materialization costs, our kernels yield much higher throughput with TwELL compared to past ELL formats.
>
> Second, since our formats are not inherently limited to GEMM, we wrote a [sub-tiled extension of our ungated kernel for GEMV](https://anonymous.4open.science/r/sparse-anon-reb-5575/code_listings/matmul_t2d.cu). Thus, we reproduced the results in Fig 3 of the TEAL paper (comparing the DejaVu and TEAL kernels at different sparsity levels) and added TwELL results:
>
> |GEMV latency(µs)|50%|75%|90%|95%|99%|
> |:-|-:|-:|-:|-:|-:|
> |Theoretical|28.3|14.6|5.6|2.6|0.6|
> |Torch|55.7|55.8|55.6|55.8|55.7|
> |DejaVu|39.2|28.4|20.4|20.0|19.3|
> |TEAL|35.1|25.9|22.7|21.9|22|
> |TwELL|34.5|23.2|15.8|13.0|6.2|
>
> As summarized above, we find that TwELL can provide further GEMV speedups, especially visible with sparsity > 75%. We hope these new comparisons will help contextualize differences with prior work and our paper's contributions.
>
> **Qwen CPT**
>
> As suggested by the review, **we collected results sparsifying Qwen2.5-7B with CPT**. While we refer to our first response to the second reviewer (h3bp) for a detailed analysis, in line with [3], our results suggest that a short CPT phase can indeed recover >95% downstream accuracy:
>
> |CPT L1|Speedup(%)|Energy sav.(%)|Acc (2B)|Acc (10B)|Acc (20B)|Acc Qwen 7B|
> |-|-:|-:|-:|-:|-:|-:|
> |2e-5|9.7|8.9|54|59|61|64|
>
> **PTQ**
>
> Based on the reviewer’s feedback, **we added results to confirm that sparsity and weight quantization are orthogonal**. To show this, we quantized our sparse models with GPTQ to 8 and 4 bits, comparing perplexity, task accuracy, and non-zeros:
>
> |L1|Acc(base/8b/4b)|CE(base/8b/4b)|#NNZ(base/8b/4b)|
> |-|-|-|-|
> |0|46.4/46.5/46.5|2.236/2.236/2.251|911/911/910|
> |3e-6|47.0/47.1/47.0|2.237/2.237/2.251|241/241/241|
> |2e-5|46.1/46.1/45.2|2.295/2.295/2.322|29.4/29.4/29.3|
>
> As summarized above, quantized sparse and non-sparse models appear to behave analogously without any adversarial interaction between the techniques.
>
> **Training Sharding**
>
> Based on the reviewer’s feedback, **we validated that sparse training is compatible with different distribution strategies**. As our code uses DeepSpeed, we compared zero1/2/3 sharding strategies:
>
> |L1|Tok/ms(zero1)|Tok/ms(zero2)|Tok/ms(zero3)|
> |-|-:|-:|-:|
> |Base|31.8|31.1|30.2|
> |1e-4|38.7|37.1|36.3|
> |2e-5|37.7|35.8|34.7|
>
> As summarized above, our results confirm that training speedups from our kernels are consistent across all sharding strategies.
>
> **Sparsity and L1**
>
> We thank the reviewer for pointing us to related works about sparsity and its applications. **We added a discussion** of the L1 loss and its properties in modern transformers [4], together with past uses of sparsity for CNN compression, such as [5] and [6].
>
> [1] Song et al Prosparse: Enhancing intrinsic activation sparsity within llms
>
> [2] Vazquez et al Improving the performance of the sparse matrix vector product with GPUs
>
> [3] Mirzadeh et al Relu strikes back
>
> [4] Trauger et al Sequence length independent norm-based generalization bounds for transformers
>
> [5] Li et al Pruning filters for efficient convnets
>
> [6] Liu et al Learning efficient convolutional networks through network slimming

---

> > ### Author Rebuttal · Reviewer_2i2g · 2026-04-01
> >
> > I am grateful for the authors' comprehensive and detailed rebuttal. I fully recognize that addressing my concerns required substantial effort and dedication, and I would like to express my sincere respect for your hard work. My questions have been thoughtfully addressed, and I am pleased to raise my score to 4. I wish you all the best in your academic endeavors.

---

### Official Review · Reviewer_h3bp · 2026-03-11

**Soundness:** 3
**Presentation:** 3
**Significance:** 3
**Originality:** 2
**Overall Recommendation:** 4
**Confidence:** 1

**Summary:**

This paper studies how to make Transformer LLMs more efficient by exploiting activation sparsity in feed-forward blocks, which dominate parameter count and FLOPs. The main contributions are two custom GPU-oriented sparse execution schemes: TwELL for fused sparse inference and a hybrid sparse/dense format for training, both supported by new Triton/CUDA kernels. Empirically, the paper shows that using ReLU plus mild L1 regularization can induce very high activation sparsity with limited loss in downstream accuracy at moderate regularization levels, while yielding measurable gains in inference speed, training throughput, energy per token, and memory in several billion-parameter settings. The paper also analyzes how sparsity varies across layers, tokens, and model scales. Overall, this research focuses on a notable context: making unstructured sparsity practically useful for modern LLM systems rather than only theoretically attractive.

**Compliance With Llm Reviewing Policy:**

Affirmed.

**Key Questions For Authors:**

Refer to Weaknesses

**Limitations:**

Refer to Weaknesses

**Strengths And Weaknesses:**

**Strengths:**

The paper’s strongest aspect is its systems contribution. The proposed TwELL inference path and hybrid training format are well motivated by the mismatch between unstructured sparsity and standard GPU kernels, and the design rationale is clearly connected to tiling, fusion, and memory-access considerations. The appendix also provides unusually concrete implementation detail, which helps technical credibility and reproducibility. The empirical story is generally convincing within the tested regime. The paper evaluates multiple model sizes, several L1 levels, and reports accuracy, cross-entropy, throughput, energy, and memory.

**Weaknsess:**

The main weakness in significance is external validity. All end-to-end results are on relatively small LLMs by current standards (0.5B–2B parameters), trained from scratch on FineWeb-derived data. That is enough to demonstrate promise, but it is still unclear how well the approach would transfer to larger production-scale models, longer contexts, or post-training adaptation of strong pretrained checkpoints. The discussion acknowledges this, but the evidence remains limited. (lacks several mainstream knowledge- and reasoning-oriented benchmarks commonly used for modern LLM assessment).

the work is meaningfully novel at the kernel/format level, but less so at the modeling level: the sparsity-inducing recipe is intentionally simple (ReLU + L1), and some conceptual foundations come from prior activation-sparsity papers. That is acceptable, but the paper’s novelty rests more on practical systems realization than on a fundamentally new ML formulation.

---

> ### Author Rebuttal · Authors · 2026-03-30
>
> We thank the reviewer for their time and constructive feedback.
>
> **Breadth of Results**
>
> Based on the reviewer’s feedback, **we added two new results covering fine-tuning larger models and additional model classes** to improve the empirical evidence in our work.
>
> First, **we swapped SiLU activations with ReLU in Qwen2.5-7B and finetuned the model with mild L1 loss for 20B tokens using Nemotron pre-training data**. After training, we evaluated downstream accuracy and efficiency:
>
> |CPT L1|Speedup (%)|Energy sav.(%)|Acc(2B)|Acc(10B)|Acc(20B)|Acc Qwen 7B|
> |-|-:|-:|-:|-:|-:|-:|
> |2e-5|9.7|8.9|54|59|61|64|
>
> As summarized above, and in line with [2], our results indicate that LLMs can recover over 95% of their downstream performance after only 20B tokens. Moreover, with our kernels, the resulting model already attains non-trivial throughput speedups and energy savings, which should expectedly grow with additional fine-tuning.
>
> Second, **we collected new results extending sparse training to MoE models at both 1.5B and 2.5B parameters**:
>
> |Model|L1 loss|Acc (%, sparse / baseline)|Speedup (%)|Energy sav. (%)|
> |:-|:-|-:|-:|-:|
> |MoE-2.5B|3e-6|45.3 / 45.1|14.4|15.6|
> |MoE-1.5B|3e-6|42.5 / 42.5|12.7|11.0|
>
> We refer to our second response to the first reviewer (HeU6) for a detailed analysis of these results, which validate the compatibility of sparsity with billion-parameter MoEs.
>
> We hope these results, extending our empirical analysis with larger models, fine-tuning, and alternative model classes, can serve to further reinforce the empirical breadth of our work.
>
> **Modeling Level**
>
> While the reviewer is correct in stating that this work’s main novelty is introducing new kernels and data formats for LLMs, in Appendix C, we also examined additional strategies at the model level for dead neuron prevention. In particular, we compared our regular dense and sparse LLMs trained with simple L1 losses against two baselines with modified training methods:
> - First, an LLM with sparsity is enacted via a delayed linear warmup stage.
> - Second, an LLM with active dead neuron detection and partial reinitialization via noise injection, inspired by techniques used to reinject plasticity in continual learning settings [3].
>
> |Model|Acc(%)|Speedup(%)|Energy sav.(%)|#NNZ|
> |:-|-:|-:|-:|-:|
> |Baseline|46.4|0.0|0.0|911|
> |Sparse  (L1)|46.2|17.9|12.1|29|
> |Sparse (warmup)|45.9|1.9|0.1|108|
> |Sparse (neuron reinit.)|46.6|19.1|14.0|29|
>
> Our results show that both warmup strategies are visibly effective at preventing dead neurons ([please refer to Figure 8, which we also provide this link](https://anonymous.4open.science/r/sparse-anon-reb-5575/dead_neuron_analysis.png)). However, as summarized above, we find that linear warmups tend to have an adversarial effect on the LLM’s resulting sparsity level and performance. In contrast, we find that targeted reinitialization is surprisingly effective at preventing dead neurons and even slightly boosts overall performance while preserving sparsity levels.
>
> In addition to these results, following the reviewer’s feedback, we tried to further reinforce our modeling analysis, with **new results examining the role of different activation functions on sparsity**. In particular, we extended our investigation of non-gated LLMs and trained two additional models at different sparsity levels with 1.5B parameters, after replacing ReLUs with ReLU^2 non-linearities [4]. To support this alternative non-linearity, we wrote new custom inference kernels still leveraging our new TwELL data format in the feed-forward blocks ([for which we share code listings at this link](https://anonymous.4open.science/r/sparse-anon-reb-5575/code_listings/matmul_t2d_relu2.cu)). After training, we collected results comparing downstream accuracy, throughput speedups, and energy savings against our non-gated LLM baselines:
>
> |Model|L1 loss|Acc(%)|Speedup(%)|Energy sav.(%)|
> |:-|:-|-:|-:|-:|
> |Non-Gated|1e-04|42.1|19.7|16.0|
> |ReLU^2|1e-04|41.9|20.2|17.2|
> |Non-Gated|2e-05|46.5|11.2|8.8|
> |ReLU^2|2e-05|47.0|13.3|9.2|
>
> As summarized above, and in line with the results in [5], we find ReLU^2 to be slightly more performant than regular non-gated LLMs. Interestingly, we find that ReLU^2 models are slightly less sparse on average (~10% more non-zeros), but also that such sparsity is more uniformly distributed across different LLM layers. Thanks to this latter property, we find that our fused kernels provide marginally higher throughput gains than in regular non-gated LLMs.
>
> We hope that this analysis and additional results will complement the sparsification analysis and provide more insights into potential future directions for more efficient LLM sparsity.
>
> [1] Yang et al Qwen2.5
>
> [2] Mirzadeh et al Relu strikes back: Exploiting activation sparsity in llms
>
> [3] Ash et al On warm-starting NN training
>
> [4] So et al Searching for efficient transformers for language modeling
>
> [5] Zhang et al ReLU^2 Wins: Discovering Efficient Activations for Sparse LLMs

---

> > ### Author Rebuttal · Reviewer_h3bp · 2026-04-05
> >
> > I am grateful for the authors' comprehensive and detailed rebuttal. I am pleased to keep my score to 4. I wish you all the best in your academic endeavors.

---

> > > ### Author Response · Authors · 2026-04-05
> > >
> > > We would like to thank the reviewer again for their time and for the acknowledgement. Please do not hesitate to let us know for follow-up questions or in case there is anything we have not addressed in our original rebuttal that we can further clarify.

---

### Official Review · Reviewer_HeU6 · 2026-03-13

**Soundness:** 3
**Presentation:** 3
**Significance:** 3
**Originality:** 3
**Overall Recommendation:** 5
**Confidence:** 3

**Summary:**

The paper studies speeding up the computation of MLPs under activation sparsity. They focus on modern gated MLPs and find that with
ReLU activation on the gate, the gate activations already have some sparsity, which allows for saving flops when computing the up and down projection of the MLP. To this end, they develop kernels for both training and inference with new sparse formats that enable speedups under this sparsity. To further increase the sparsity of activations they experiment with adding an auxiliary l1 regularization loss on the activations, and find that even small decays significantly increase sparsity at little degradation.

**Compliance With Llm Reviewing Policy:**

Affirmed.

**Ethical Review Concerns:**

yes

**Final Justification:**

My concerns regarding missing baselines and MoE integration have been addressed and I believe this work should be accepted.

**Key Questions For Authors:**

Have you experimented with other linearities such as Relu^2 that may naturally induce more sparsity?

**Limitations:**

yes

**Strengths And Weaknesses:**

Strengths:
- The speedups that are achieved for high sparsities are significant (~30%)
- The idea behind the kernels, as well as the kernels themselves are of value independently of the proposed l1-regularization that induces sparsity. Future work might come up with better method to induce sparsity, or with a different architecture that's naturally more sparse.
- Both training and inference is considered, and the additional challenges of training (backpropagation) are addressed.
- Kernels and code are released. The paper is very well written.
- The analysis of sparsity distribution across model depth under a fixed l1 regularization penalty is interesting. L1-regularization seems to be a natural way to enforce something similar to non-uniform compression.

Weaknesses:
- The paper should include a more extensive related work section on kernels for sparsity. For example, MACKO [1] should be discussed.
- Only dense MLPs are studied, however, transformers widely use MoE layers, and the activated experts could potentially have a different sparsity pattern than dense MLPs.
- While the proposed l1 regularization does allow for speedups, it is not clear if the speedups outweigh the degradation in token efficiency.


[1] Macko, V., & Boza, V. (2025). MACKO: Sparse Matrix-Vector Multiplication for Low Sparsity. arXiv:2511.13061.

---

> ### Author Rebuttal · Authors · 2026-03-30
>
> We thank the reviewer for their time and constructive feedback, based on which we made several extensions to improve our work.
>
> **Related Work Extensions**
>
> We thank the reviewer for pointing us to MACKO [1]. Following their feedback, **we have extended the related works section with more references to prior work trying to leverage sparsity for efficient LLM inference***, including MACKO, DejaVu [2], TEAL [3], and ProSparse [4]. Moreover, we also added a discussion to explain how the kernels and data formats introduced in these prior papers focus on memory-bound GEMV operation, taking a much different approach to sparsity than our kernels, which instead focus on compute-bound GEMM operations for both batched training and inference.
>
> In case the reviewer is interested, we note that **we also added two new direct comparisons against TEAL, ProSparse, and DejaVu**, which empirically show the limits of prior sparse kernel designs in the batched GEMM setting, and even how our TwELL data format can also be used to accelerate GEMV operations. We refer to our first response to reviewer 2i2g below for more details and analysis of these new results.
>
> **Extending Sparse Training and Kernels to MoEs**
>
> Following the reviewer's comments, **we also collected new results extending sparse training to MoE models at both 1.5B and 2.5B parameters**. We note that we re-tuned the L1 coefficient and found that MoE models need roughly 10x lower L1 losses to achieve similar numbers of non-zeros than regular dense LLMs. After training, we compared downstream accuracy, speedups, and memory savings after applying our inference kernels leveraging our new TwELL data format in every expert:
>
> |Model|L1 loss|Acc (%, sparse / baseline)|Speedup (%)|Energy sav. (%)|
> |:-|:-|-:|-:|-:|
> |MoE-2.5B|3e-6|45.3 / 45.1|14.4|15.6|
> |MoE-1.5B|3e-6|42.5 / 42.5|12.7|11.0|
>
> As summarized above, these new results are generally consistent with our prior results on regular dense LLMs, showing how kernels can provide visible speedups and energy savings, and empirically illustrating the complementarity of structured and unstructured sparsity.
>
> **ReLU^2 Models and Kernels**
>
> Following the reviewer’s question, **we trained new non-gated LLMs with 1.5B parameters and ReLU^2 non-linearities at two sparsity levels**. Furthermore, we wrote new custom inference kernels leveraging our new TwELL data format to support ReLU^2 non-linearities in the feed-forward blocks ([for which we share code listings at this link](https://anonymous.4open.science/r/sparse-anon-reb-5575/code_listings/matmul_t2d_relu2.cu)). After training, we collected results comparing downstream accuracy, throughput speedups, and energy savings against our non-gated LLM baselines:
>
> |Model|L1 loss|Acc (%)|Speedup (%)|Energy sav. (%)|
> |:-|:-|-:|-:|-:|
> |Non-Gated|1e-04|42.1|19.7|16.0|
> |ReLU^2|1e-04|41.9|20.2|17.2|
> |Non-Gated|2e-05|46.5|11.2|8.8|
> |ReLU^2|2e-05|47.0|13.3|9.2|
>
> As summarized above, and in line with the results in prior work [5], we find ReLU^2 to be slightly more performant than regular non-gated LLMs. Interestingly, we find that ReLU^2 models are slightly less sparse on average (~10% more non-zeros), but also that such sparsity is more uniformly distributed across different LLM layers. Thanks to this latter property, we find that our fused kernels provide marginally higher throughput gains than in regular non-gated LLMs. We hope that these additional results will provide more insights into the role of activation functions and highlight future directions for more efficient LLM sparsity.
>
> [1] Macko et al MACKO: Sparse Matrix-Vector Multiplication for Low Sparsity
>
> [2] Liu et al Deja vu: Contextual sparsity for efficient llms at inference time
>
> [3] Liu et al Training-free activation sparsity in large language models
>
> [4] Song et al Prosparse: Introducing and enhancing intrinsic activation sparsity within large language models
>
> [5] Zhang et al ReLU^2 Wins: Discovering Efficient Activation Functions for Sparse LLMs

---

> > ### Author Rebuttal · Reviewer_HeU6 · 2026-04-01
> >
> > My concerns regarding missing baselines and MoE integration have been addressed and I believe this work should be accepted.

---

### Decision · Program_Chairs · 2026-04-30

**Decision:**

Accept (regular)

**Comment:**

This paper presents a practical method for leveraging unstructured sparsity in LLMs, specifically targeting feed-forward layers. The primary contribution is systems-oriented: the introduction of novel sparse data formats and custom CUDA/Triton kernels optimized for both training and inference, paired with a ReLU+L1 sparsification strategy. Results demonstrate clear improvements in efficiency with minimal performance trade-offs.

The submission received consistent support from all four reviewers, with final scores of 5, 4, 4, and 4 (average 4.25). Reviewers praised the practical systems contributions, hardware-aware design, and strong reproducibility. While initial concerns were raised regarding evaluation breadth and the lack of mainstream baselines, the authors effectively addressed these during the rebuttal. Specifically, the inclusion of Qwen2.5-7B results, MoE experiments, and PTQ compatibility satisfied the reviewers, leading to a consensus for acceptance. The paper offers a timely solution for LLM efficiency. Accordingly, we recommend Accept.